# NH₃-promoted hydrolysis of NO₂ induces explosive growth in HONO

**Wanyun Xu[1], Ye Kuang[2,\*], Chunsheng Zhao[3], Jiangchuan Tao[2], Gang Zhao[3], Yuxuan Bian[4], Wen Yang[5], Yingli Yu[3], Chuanyang Shen[3], Linlin Liang[1], Gen Zhang[1], Weili Lin[6], Xiaobin Xu[1]**

[1] State Key Laboratory of Severe Weather, Key Laboratory for Atmospheric Chemistry, Institute of Atmospheric Composition, Chinese Academy of Meteorological Sciences, Beijing, 100081, China.

2 Institute for Environmental and Climate Research, Jinan University, Guangzhou, China.

[3] Department of Atmospheric and Oceanic Sciences, School of Physics, Peking University, Beijing, China

[4] State Key Laboratory of Severe Weather, Chinese Academy of Meteorological Sciences, Beijing, 100081, China

[5] State Key Laboratory of Environmental Criteria and Risk Assessment, Chinese Research Academy of Environmental Sciences, Beijing, 100081, China

[6] College of Life and Environmental Sciences, Minzu University of China, Beijing, 100081, China

Corresponding author: Ye Kuang (kuangye@jnu.edu.cn)

**Abstract**

The study of atmospheric nitrous acid (HONO), which is the primary source of OH radicals, is crucial to atmospheric photochemistry and heterogeneous chemical processes. The heterogeneous NO₂ chemistry under haze conditions was pointed out to be one of the missing sources of HONO on the North China Plain, producing sulfate and nitrate in the process. However, controversy exists between various proposed mechanisms, mainly debating on whether SO₂ directly takes part in the HONO production process and what roles NH₃ and the pH value play in it. In this paper, never before seen explosive HONO production was reported and evidence was found for the first time in field measurements during fog episodes (usually with 4<pH<6) and haze episodes under high relative humidity (pH≈4), that NH₃ was the key factor that promoted the hydrolysis of NO₂, leading to explosive growth of HONO and nitrate under both high and lower pH conditions. The results also suggest that SO₂ takes minor or insignificant part in the HONO formation during fog and haze events, but was indirectly oxidized upon the photolysis of HONO through subsequent radical mechanisms. Aerosol hygroscopicity significantly increased with the rapid inorganic secondary aerosol formation further promoting the HONO production as a positive feedback. For future photochemical and aerosol pollution abatement, it is crucial to introduce effective NH₃ emission

control measures, since the $NH_3$-promoted $NO_2$ hydrolysis is a large daytime HONO source,
releasing large amounts of OH radicals upon photolysis, which will contribute largely to both
atmospheric photochemistry and secondary aerosol formation.

## 1 Introduction

Nitrous acid (HONO) plays a vital role in atmospheric chemistry due to the fact that its
photolysis is a major source (Michoud et al., 2014;Kleffmann et al., 2005) of hydroxyl radical (OH)
which determines the atmospheric oxidative capacity and plays crucial role in tropospheric
chemistry in processes such as the ozone formation, the degradation of volatile organic compounds
and the  secondary aerosol formation (Cheng et al., 2016;Wang et al., 2016).  Hence, the source
study of nitrous acid (HONO) is of crucial importance for the understanding of the tropospheric
chemistry, for chemistry and climate modelling and for developing effective pollution control
strategies (Lu et al., 2018).
The North China Plain (NCP) is troubled by the persistent complex air pollution with high
loadings of both photochemical pollutants and particulate pollution (Zheng et al., 2015;Ran et al.,
2011) and the simultaneous mitigation of the two types of pollution has encountered trouble due
to the nonlinear dependence of ozone on NOx (Xing et al., 2018). Unknown daytime sources of
HONO caught attention during the past few years (Michoud et al., 2014;Liu et al., 2014;Su et al.,
2011) and results from a recent study indicate that an additional missing source is required to
explain more than 50% of observed HONO concentration in the daytime in Western China (Huang
et al., 2017). Results from several recent studies demonstrate that intense heterogeneous
conversion of $NO_2$ to HONO on particle surfaces might be a significant source of HONO (Cui et
al., 2018;Liu et al., 2014).
Two main HONO heterogeneous production pathways involving aerosol water and $NO_2$
were proposed. In light of drastic decrease of solar radiation during severe haze events and rich
ammonia conditions on the NCP, the first pathway hypothesized that $NO_2$ (g) dissolved in aerosol
water at aerosol pH > 5.5 rapidly formed HONO while oxidizing $HSO_3^-$ (aq) to sulfate. The
stoichiometry of this mechanism is as follows (Cheng et al., 2016;Wang et al., 2016):

$2NO_2$ (aq) + $HSO_3^-$ (aq) + $H_2O$ (l) ➜ $2H^+$ + $HSO_4^-$ (aq) + $2NO_2^-$ (aq).   (R1)

Based on this mechanism, good agreement between modelled and observed sulfate
formation rates were achieved. However, the assumption that the pH of ambient aerosols can reach
beyond 5.5 is a debatable issue. Results from several most recent studies indicate that the pH of
ambient aerosols fall in the range of 3-5 in most cases (Ding et al., 2018;Liu et al., 2017a;Song et
al., 2018). Given this, it was proposed that HONO and $NO_2^-$ were produced in the hydrolysis
process of $NO_2$, releasing OH radicals upon photolysis, which indirectly oxidize $SO_2$ to sulfate (Li
et al., 2018b):

$2NO_2$ (g) + $H_2O$ (l) ➔ $H^+$ + $NO_3^-$ (aq) + HONO.  (R2)

Results of Yabushita et al. (2009) suggest that anions (such as $Cl^-$, $Br^-$ and $I^-$) greatly
enhance the hydrolysis of $NO_2$ on water, and the $NO_2$ uptake coefficients of R2 can be enhanced
several orders of magnitude by increasing electrolyte concentration. The ambient aerosol particles
in the boundary layer are in aqueous phase under high RH (Liu et al., 2017b) and the aerosol or
fog water is not pure with different dissolved anions (Wu et al., 2018;Lu et al., 2010). Therefore,
HONO and nitrate formed through this mechanism should be independent of aerosol acidity, and
should be primarily affected by the aerosol surface area density ($S_a$), aerosol liquid water content
and $NO_2$ concentration (Li et al., 2018b). Moreover, recent theoretical simulations have proposed
a HONO formation mechanism involving $NO_2$ and water and have identified that $NH_3$ can promote
the hydrolysis of $NO_2$ (Li et al., 2018a) (R2). Despite of this, no direct evidence from field
observations were available in this paper to support their findings.
Although the proposed HONO formation mechanisms are all heterogeneous reactions of
$NO_2$, the details of how $SO_2$, pH and $NH_3$ are involved in heterogeneous formation are still under
debate (Li et al., 2018b) and a clear mechanism is still missing in current models to explain both
the daytime concentration of observed HONO and the secondary inorganic aerosol formation.
Measurements of HONO are rare and simultaneous observations of HONO and aerosol physical
and chemical characteristics are lacking to thoroughly analyze or directly support the aerosol
heterogeneous HONO formation mechanisms involving $NO_2$. In this paper, we present f
simultaneous measurements of HONO, sulfate and nitrate as well as other precursor gases,
oxidants and meteorological parameters during both fog and haze episodes under high ambient
RH. Fog water pH is usually greater than 5.5 in eastern China (Safai et al., 2008;Lu et al., 2010),
while calculations in this work and previous studies collectively indicate a moderately acidic

condition (4<pH<5) for fine particles in northern China winter haze. The observational results unveil that $NH_3$ is the key factor that promotes the hydrolysis of $NO_2$, resulting in explosive formation of HONO, nitrate and sulfate.

## 2 Site description and instruments

From $15^{th}$ Oct. to $25^{th}$ Nov. 2016, a field campaign intended to study sulfate formation was conducted at the Ecological and Agricultural Meteorology Station ($39°09'N, 115°44'E$) of the Chinese Academy of Meteorological Sciences. The site is partly composed of experimental farmland and is also surrounded by farmland and small residential towns (nearest town ~1.5 km). It is located between Beijing (~ 100km) and Baoding (~40km), two megacities on the North China Plain (Fig. 1). During this field campaign, an In situ Gas and Aerosol Compositions Monitor (IGAC, Fortelice International Co.,Taiwan) was used for monitoring water-soluble ions ($Na^+$, $K^+$, $Ca^{2+}$, $Mg^{2+}$, $NH_4^+$, $SO_4^{2-}$, $NO_3^-$,$NO_2^-$, $Cl^-$) of $PM_{2.5}$ (particulate matter with aerodynamic diameter less than 2.5 µm) and trace gases including HONO, $SO_2$, $NH_3$, HCl, and $HNO_3$ with a time resolution of 1h. The IGAC system draws in ambient air through a PM10 inlet and passes the sample through a sharp-cut PM2.5 cyclone at a flowrate of 16.7 L $min^{-1}$. The total length of the stainless steel sampling line is approximately 2 m, with an inner diameter of 3.18 cm (1.25 inch), resulting in a residence time below 6 s, suggesting that underestimates in $NH_3$ possibly caused by adsorption on the stainless steel sampling tube as was proposed by Young et al. (2016) might be unimportant. A vertical annular denuder wetted with dilute $H_2O_2$ solution ($5x10^{-3}$ M) collects the trace gases and converts $SO_2$ rapidly to $SO_4^{2-}$, preventing $SO_2$ from reacting with $NO_2$ in the absorption solution to produce HONO artefacts. A scrub and impact aerosol collector under the denuder is mounted at an inclined angle to capture particles based on impaction after condensation growth. Two separate Ion Chromatographs are used to respectively analyze anions and cations for the gas and aerosol liquid extracts which were injected from the denuder and the aerosol collector once an hour. The detection limits are below 0.12 μg $m^{-3}$ and the background concentration of most water-soluble inorganic ions within the instrument were below 0.11 μg $m^{-3}$, only with $SO_4^{2-}$ showing a background concentration of 1.10 μg $m^{-3}$ (Young et al., 2016). Considering the severe pollution state the NCP is under, these measurement uncertainties are fully acceptable. The instrument has shown good performance in the past, agreeing well with filter based samples (Liu et al., 2017a). Standard LiBr solution was continuously added to the aerosol liquid extracts during

the measurements, to ensure the sampling and analyzing process is stable. The swing amplitude
was within the range of three standard deviation, confirming the stability of the ion analyzing
system throughout the campaign. A mixed standard solution was diluted to perform multipoint
calibrations (at 5, 10, 20, 50, 100, 200, 500 and 1000 ppb concentrations) at the beginning and at
the end of the campaign for the ions $Na^+$, $K^+$, $Ca^{2+}$, $Mg^{2+}$, $NH_4^+$, $Li^+$, $SO_4^{2-}$, $NO_3^-$, $NO_2^-$, $Cl^-$, $Br^-$,
with the $R^2$ of the calibrations reaching above 0.9999. A comparison between $NH_3$ observed by
IGAC and by an economical $NH_3$ analyser (LGR, DLT-100, details see Meng et al. (2018)) yielded
an overall slope of 0.91 with R=0.63 (Fig.S1a). A better comparison result (slope of 1.03, R=0.74)
would be obtained if data associated with RH≥80 were excluded (Fig.S1b). The overestimation of
LGR instruments compared to denuder based instruments has also been reported in Teng et al.
(2017), suggesting possible interference of water vapor on $NH_3$ measurements. As can be seen in
Fig.S2, both instruments captured the same the diurnal variation of $NH_3$ during the four case
episodes in this study, which proves that the IGAC instrument was able to capture the overall
variation trends of $NH_3$. Since both instruments have their uncertainties, we decided to use the
$NH_3$ measured by the IGAC instrument for better consistency with the other data.

NOx and CO were observed using commercial instruments from Thermo Electronics

(Model 42CTL and 48CTL), while the Aerolaser AL2021 $H_2O_2$-monitor was used to measure
$H_2O_2$ concentrations. The $NO_x$ instrument uses a Mo-based converter, which would result in
interference of $NO_z$ species (e.g. HONO, $HNO_3$, PAN, etc.) on $NO_2$. Here, we define
$NO_2^*$=$NO_{2,meas}$ -HONO-$HNO_3$, and use it to approximate the true $NO_2$ concentration. The ambient
RH, temperature, wind speed and wind direction were observed using an automatic weather station.
The dry state particle number size distributions (PNSDs) in the diameter range of 3nm to 10μm,
were jointly measured by a scanning mobility particle size spectrometer (SMPS) and an
Aerodynamic Particle Sizer (APS, TSI Inc., Model 3321). The ambient aerosol liquid water
concentrations were calculated based on measurements of a three-wavelength humidified
nephelometer system (Kuang et al., 2018). The aerosol hygroscopicity parameter κ (Petters and
Kreidenweis, 2007) is calculated using the method proposed by Kuang et al. (2017). The aerosol
surface area concentration ($S_A$) is calculated based on measured PNSD and the retrieved
hygroscopicity parameter κ based on measurements of the humidified nephelometer system.

## 3 Observed simultaneous rapid increase of HONO, nitrate and sulfate

The time series of HONO, sulfate, nitrate and ammonium and precursor gases, meteorological parameters and other parameters are shown in Fig. 2. During this observation period, HONO concentration ranged from 0.31 to 17.6 ppb (ranged from 0.3 to 6.0 ppb during most periods) with an average of 3.0 ppb. $NO_2^*$ concentration ranged from 7.1 to 56.3 ppb with an average of 28.9 ppb. $NH_3$ concentration ranged from 0.05 to 30 ppb with an average of 12.3 ppb. The $HONO/NO_2^*$ ratio ranged from 0.02 to 0.6 with an average of 0.11, which is moderately higher than the previously reported results in Eastern China (Cui et al., 2018;Liu et al., 2014). This is because compared to their measurements, the $PM_{2.5}$ mass concentrations and $S_a$ in this study are much higher (As shown in Fig.3). Additionally, $HONO/NO_2^*$ increases with $PM_{2.5}$ and $S_a$, which is consistent with previous results, suggesting that aerosol might have promoted the conversion from $NO_2$ to HONO. In Cui et al. (2018), under the observed $PM_{2.5}$ range of 0 to 100 $\mu g\ m^{-3}$, $HONO/NO_2$ ranged from 0.0013 to 0.17, with an average of 0.062, while in this study the average $HONO/NO_2^*$ also increases from 0.06 to 0.07 for the same $PM_{2.5}$ range. Liu et al. (2014) reported that average $HONO/NO_2$ increased from 0.04 to 0.1 when $S_a$ increased from 200 to 1100 $\mu m^2\ cm^{-3}$, while, the average $HONO/NO_2^*$ in this study increased from 0.05 to 0.15 for the same $S_A$ range. The comparison suggests that our HONO measurements were comparable to those made using other instruments previously reported in Eastern China. Further, it can be noticed that for the relatively lower $PM_{2.5}$ concentration and $S_A$ range, $HONO/NO_2^*$ increased rapidly with increasing aerosol loading, while after a critical concentration ($PM_{2.5}>225\ \mu g\ m^{-3}$, $S_a>1100\ \mu m^2\ cm^{-3}$) the increase came to a halt. This indicates that under relatively cleaner conditions, the heterogeneous conversion of $NO_2$ to HONO might have been limited by aerosol surface area density. However, under severe haze pollution or foggy conditions with sufficient $S_a$ available for heterogeneous reactions, the HONO formation was not sensitive to the change in $S_a$ anymore.

Four rapid HONO formation events were identified in Fig.2, two under foggy conditions and the other two under severe haze with high RH conditions. In the following Sect. 3.1 and 3.2, the variations of the pollutants connected to HONO formation during the four cases will be described in detail, so that the mechanism behind such rapid HONO production under conditions when HONO formation was not sensitive to $S_A$ can be better discussed in Sect. 4.

### 3.1 Explosive growth of HONO during fog episodes

Two dense fog episodes with rapid HONO increase were observed for the first time in China, occurring on the 4th and 5th Nov. 2016. From satellite images (Fig. 1) it can be seen that on the 5th Nov., a wide area of the NCP was shrouded by fog before noon (about 11:30) including the observation site, however, the fog area reduced in the afternoon (about 13:30) and dissipated near the observation site. The evolution of the fog-shrouded area during these two days was also observed by a geostationary satellite (http://www.eorc.jaxa.jp/ptree/index.html). These two fog episodes offer us a great opportunity to study the hydrolysis process of $NO_2$ (R2) and the role of $SO_2$ in heterogeneous HONO production in fog water (R1), which usually show pH above 5.5 (Safai et al., 2008;Lu et al., 2010).

The time series of simultaneously observed meteorological parameters, concentrations of nitrate, ammonium, sulfate and their precursor gases $SO_2$, $NO_2^*$, NO and $NH_3$, as well as atmospheric oxidants such as $O_3$, $H_2O_2$ and other parameters including CO, which is indicative of transport processes during the two days with fog episodes are shown in Fig. 4. From 0:00 (Beijing local time) on the 4th Nov., the ambient RH continuously increased and reached 100% near 5:00, and lasted about 8.5 hours before it dropped below 100% near 13:30. However, at 15:30, the ambient RH began to rise again and reached 100% near 19:30, and then sustained until 12:00 on the 5th Nov. The latter fog episode lasted about 18.5 hours.

During the first fog episode, the rapid increases of HONO, nitrate, sulfate and ammonium were observed from 8:50 to 11:30 (Case1). HONO increased from 3.6 ppb to 10.6 ppb, with the most rapid increase occurring around 11:00 at a rate of 5.5 ppb $h^{-1}$. During the HONO increasing period, the variation characteristics of related trace gases and other parameters are as follows. $NH_3$ concentration increased slowly at first and then increased drastically near 11 am (10 ppb $h^{-1}$). $SO_2$ concentration remained almost constant at first and then increased from near 0.25 ppb to 0.4 ppb. $NO_2^*$ concentration varied little but decreased when HONO was increasing, while NO concentration increased first and then decreased. $H_2O_2$ concentration is continuously increasing, but $O_3$ concentration remained near zero. CO concentration remained almost constant (~2.5 ppm), suggesting that there was no evident plume transport during this process. Wind speed was less than 2 m $s^{-1}$, and dropped almost to 0 m $s^{-1}$ when HONO concentration dramatically increased, further supporting the fact that the drastic increase was not caused by transport processes. Ammonium, nitrate and sulfate concentration steadily increased from 7.5, 13.2, 13.7 μg $m^{-3}$ to 14.3, 30.4,

$31.0 \, \mu g \, m^{-3}$, respectively. A noticeable increase in nitrite was also observed, when HONO
increased most rapidly. It should be noted that the cutting diameter of the IGAC instrument is 2.5
μm, which means that observed concentrations only represent the variation of inorganics ions in
aerosol water, and that of fog droplets were not included.
During the second fog episode, HONO, nitrate, sulfate and ammonium started to increase
rapidly from 9:30 and reached a plateau near 12:30, when the fog started to dissipate (Case2).
HONO increased from 3 ppb to 9.5 ppb, with the fastest increase occurring near 11:00 at a rate of
$3.5 \, ppb \, h^{-1}$. $NH_3$ concentration increased steadily from 5 ppb to 24 ppb. $SO_2$ concentration
increased steadily from 0.25 ppb to 1.25 ppb. $NO_2^*$ concentration decreased continuously at the
very beginning (near 40 ppb) and then increased slightly, while NO concentration remained almost
constant (near 30 ppb) throughout the entire fog period. $H_2O_2$ concentration increased slightly at
first and then rose rapidly towards the end of the fog period. $O_3$ concentration increased very
slightly. CO concentration remained also near constant (~3 ppm). Wind speed was steadily below
$2 \, m \, s^{-1}$ at the beginning, however, began to increase quickly at noon. Ammonium, nitrate and
sulfate concentration grew steadily from 8.1, 17, 3.8 $\mu g \, m^{-3}$ to 15.3, 39.3, 8.0 $\mu g \, m^{-3}$, respectively.
The variation of nitrite was very similar to that of HONO. The variation of wind speed
demonstrated that at the very beginning of the HONO increase, the air mass was relatively stagnant,
but became more turbulent upon the fog dissipation.

**3.2 Explosive growth of HONO during haze episodes with high RH conditions**

Two episodes with rapid HONO increase under severe haze with high RH conditions
occurred on the 11[th] and 14[th] Nov., respectively. The time series of simultaneously observed
meteorological parameters, concentrations of nitrate, ammonium, sulfate and their precursor gases
$SO_2$, $NO_2^*$, NO and $NH_3$, as well as oxidants including $O_3$, $H_2O_2$ and other parameters such as CO
concentration, aerosol volume concentration in dry state and aerosol liquid water content during
the two days are shown in Fig.5.
On the 11[th] Nov., HONO started rising from 6:30 (3.4 ppb) and came to a halt at 9:00 (11.5
ppb) (Case 3). The quickest increase of HONO occurred near 9 o'clock with a rate of $5.6 \, ppb \, h^{-1}$.
The ambient RH decreased rapidly (from foggy condition to near 75%). $NH_3$ increased slowly at
first and then grew rapidly. $NO_2^*$ decreased slowly and $SO_2$ remained low. The total volume
concentration of $PM_{2.5}$ was decreasing. Ammonium, nitrate and sulfate concentrations increased
very slowly at first and then evident increase was observed in ammonium and nitrate. The decrease
in dry state volume concentration of $PM_{2.5}$ demonstrate that the air mass is not quite steady due to
transport or boundary layer processes. The slight increase of nitrate and sulfate despite the drop in
total $PM_{2.5}$ concentration suggest that the nitrate and sulfate produced during the increasing process
of HONO outgrew those lost to boundary layer mixing and transport.

On the 14[th] Nov., HONO increased drastically near 11:00, reaching 17.6 ppb at 11:30 (16.1
ppb h[-1]) and then dropped promptly to 4 ppb at 12:30 (Case 4). This phenomenon took place when
the fog dissipated and the ambient RH abruptly dropped to near 85%. Key variation features of
other parameters are as follows. $NH_3$ increased rapidly from 9.7 ppb to 30 ppb. $NO_2^*$ concentration
was decreasing when HONO quickly increased, while $SO_2$ concentration remained low. The
concentration of sulfate and nitrate also increased quickly. Volume concentration of $PM_{2.5}$ was
decreasing, indicating that even more sulfate and nitrate were formed than the observed growth in
their concentrations. The photolysis of HONO was high probably the cause for its drastic decrease.
Note that the HONO was not increasing during the period where only $NO_2^*$ increased rapidly and
$NH_3$ varied little.

**4 Discussions**

**4.1 HONO budget analysis**

In these four rapid HONO increasing episodes, the maximum HONO growth rates
(d[HONO]/dt) all exceed 5 ppb h[-1], and even reach beyond 16 ppb h[-1]. Such high HONO growth
rates as observed in this study were not yet reported in literature. In this section, we perform a
budget analysis by estimating the net HONO production accounting for currently known sources
and sinks and by comparing it to observed dHONO/dt. Thereby we can discuss whether the
observed HONO formation events can be explained by currently known mechanisms and try to
identify which mechanisms are determining the variation of HONO.

The net HONO production rate can be estimated by accounting for all the currently known
sources and sinks using the following equation (Huang et al., 2017;Zhang et al., 2019) :
$$P_{HONO}^{net} = P_{emi} + P_{hom}^{net} + P_{het} - L_{pho} - L_{dep}, \qquad \text{(Eq.1)}$$
where $P_{emi}$ is the total emission rate of HONO, $P_{hom}^{net}$ the net HONO production in homogenous
gas phase reactions, $P_{het}$ the HONO produced via heterogeneous reactions, $L_{pho}$ the loss of HONO
due to photolysis and $L_{dep}$ the loss of HONO due to deposition.
Previous studies have shown that HONO can be emitted through biomass burning and
vehicles (Nie et al., 2015;Huang et al., 2017). Biomass burning contributes to HONO mainly by
increasing $S_a$ and $NO_2$ conversion efficiency (Nie et al., 2015). Under foggy conditions, surface
area was not the limiting factor to the $NO_2$ conversion. During the haze events, $S_a$ was decreasing
due to decreasing humidity and aerosol water content. Hence, the variation of surface area cannot
explain the observed HONO increases. According to the mapped fire spots on the days of the
HONO events (Fig.S3), there was no fire within 20 km distance to the site. $K^+$ is often used as an
indicator for biomass burning. The average $K^+$ concentration during the whole campaign ranged
from 0.022 to 5.95 µg m$^{-3}$, with an average of 1.28 µg m$^{-3}$. The $K^+$ level during the four events
were 1.39, 1.08, 1.51 and 1.54 µg m$^{-3}$, respectively, showing no evident sign of biomass burning.
Hence, only vehicle emissions were considered in this study.
Vehicle emissions can be estimated using the following equation:
$P_{vehicle} = R_{emission} \times [NO_x]_{vehicle},$ (Eq.2)
where $R_{emission}$ is the vehicle emission ratio and $[NO_x]_{vehicle}$ the $NO_x$ concentration from vehicle
emissions. The $NO/NO_x$ ratio during the HONO increasing episodes ranged from 0.37 to 0.76,
suggesting that the air masses were relatively aged compared to freshly emitted air mass from
exhaust ($NO/NO_x>0.9$). Here, $P_{vehicle}$ is estimated assuming all the measured $NO_x$ came from
vehicle emissions and an emission ratio of 1%, which is higher than the upper limit of 0.8% used
in Huang et al. (2017), to obtain an upper limit for vehicle emissions.
HONO can be formed in gas phase reactions of NO with OH radicals and is lost through
direct reactions with OH radicals. The net production of HONO via homogeneous reactions can
be estimated using the equation:
$P_{hom}^{net} = k_{NO+OH}[NO][OH] - k_{HONO+OH}[HONO][OH],$ (Eq. 3)
where $k_{NO+OH}$ ($7.2 \times 10^{-12}$ cm$^{-3}$ s$^{-1}$) and $k_{HONO+OH}$ ($5.0 \times 10^{-12}$ cm$^{-3}$ s$^{-1}$) are the rate constants of
the reactions of NO and HONO with OH, at 298 k, respectively (Li et al., 2012). The diurnal
variation of OH concentrations was inferred from Whalley et al. (2015), replacing OH under fog
conditions with $1\times10^5$ cm$^{-3}$ (Fig.S4).
Heterogeneous conversion of $NO_2$ on aerosol and ground surface is considered a major
source for HONO. However, the detailed mechanism (R1 or R2?) is still under debate and different
studies have shown a large variability in the range of estimated $NO_2$ uptake coefficient. Typically,
the conversion of $NO_2$ on aerosol and ground surface is parameterized as a linear function of $NO_2$
uptake coefficients and surface to volume ratios (or $S_a$) (Xue et al., 2014;Li et al., 2018b):
$P_{het} = (k_g + k_a)[NO_2^*],$     (Eq.4-1)
$k_g = \frac{1}{8} \cdot \vartheta_{NO2} \cdot \gamma_g \cdot \frac{S}{V},$     (Eq.4-2)
$k_a = \frac{1}{4} \cdot \vartheta_{NO2} \cdot \gamma_a \cdot S_a,$     (Eq.4-3)
where $\vartheta_{NO2}$ stands for the mean molecular speed, $\gamma_g$ and $\gamma_a$ for the uptake coefficient on ground
and aerosol surface, S/V for the surface to volume ratio and $S_a$ for the ambient aerosol surface area
density. For $NO_2$ conversion on ground surface, $\gamma_g$ is assumed to be $1\times10^{-6}$ and S/V is assumed
to be 0.1 $m^{-1}$ (Li et al., 2010;Xue et al., 2014;Vogel et al., 2003). Since no measurements of fog
droplet surface areas were made in this experiment, estimates for $NO_2$ conversion under foggy
conditions could not be incorporated. For non-fog conditions, the ambient $S_a$ calculated using the
simultaneously measured PNSD and aerosol hygroscopicity parameter derived from
measurements of a humidified nephelometer system were applied to further calculate the variation
of the HONO production on aerosol surface. Here, with an overall consideration of the $\gamma_a$ used in
past literature (Li et al., 2010;Xue et al., 2014;Li et al., 2018b), $\gamma_a$ was assumed to be $5\times10^{-6}$, $2\times10^{-4}$
and $2\times10^{-4}\times$(solar radiation/400) for nighttime, daytime with solar radiation below and above
400 $Wm^{-2}$, respectively, to account for both anion-enhanced and photo-enhanced $NO_2$ conversion.

319        HONO loss through photolysis reactions were calculated as:

$L_{pho} = J_{HONO}[HONO],$     (Eq.5)
where JHONO was modelled using the TUV radiative transfer model (version 5.3,
http://www2.acom.ucar.edu/modeling/tuv). The required single scattering albedo and aerosol
angstrom exponent were estimated using simultaneously measured PNSD and BC measurements
(Kuang et al., 2015), while the 550nm aerosol optical depth (AOD) was assumed to vary with RH
(Table S1).

326        Loss through dry deposition was estimated using equation 6:

$L_{dep} = \frac{v_{dep}}{H}[HONO],$     (Eq.6)
where the dry deposition rate $v_{dep}$ was assumed to be 0.3 cm $s^{-1}$ according to (Stutz et al., 2002)
and the boundary layer height H was interpolated from ECWMF ERA-interim data
(http://apps.ecmwf.int/datasets/data/interim-full-daily/).
The comparison between the calculated HONO net production rate and actually measured
HONO variation rate (d[HONO]/dt) is displayed in Fig. 6. The estimated upper limit for vehicle
emissions displays little variability during the day, with slight decreasing trends during the four
events, proving that the observed HONO production could not have been caused by direct vehicle
emissions. The net gaseous phase production of HONO ($P_{hom}^{net}$) contributed 0.15-0.18, 0.04-0.07,
0.27-1.04 and 0.25-1.53 ppb h$^{-1}$ during the 4 case events, displaying little influence during fog
events and more during haze events. However, the estimated $P_{hom}^{net}$ was far from sufficient to
explain the observed d[HONO]/dt. Dry deposition was typically high during the night within the
shallow nocturnal boundary layer and decreased during the day with the increase of the boundary
layer height. The calculated $L_{dep}$ contributed 0.5-0.9, 0.4-0.6, 2.7-4.3 and 0.05-0.3 ppb h$^{-1}$ to the
loss of HONO. No significant decreases in $L_{dep}$ were observed during the two fog events, while
increases were detected during the cases on 11$^{th}$ and 14$^{th}$ Nov. Not only was the variation in $L_{dep}$
unable to explain observed HONO productions, it further added to the discrepancy between
observed and calculated d[HONO]/dt. During the four case events the $J_{HONO}$ respectively increased
from $0.7\times10^{-4}$ to $2.5\times10^{-4}$ s$^{-1}$, $1.6\times10^{-4}$ to $2.4\times10^{-4}$ s$^{-1}$, $0.03\times10^{-4}$ to $1.4\times10^{-4}$ s$^{-1}$ and $1.6\times10^{-4}$ to
$4.4\times10^{-4}$ s$^{-1}$, with $L_{pho}$ contributing 0.9-8.9, 2.2-7.8, 0.03-5.5 and 0.8-26.4 ppb h$^{-1}$ to the loss of
HONO. $J_{HONO}$  increased significantly by the end of the HONO growth events to $2.9\times10^{-4}$, $4.3\times10^{-4}$,
$2.6\times10^{-4}$ and $6.6\times10^{-4}$ s$^{-1}$, respectively, suggesting  that the rapid drop of HONO concentrations
was high probably caused by the rapid photolysis. Overall, $L_{pho}$ contributed most to the discrepancy
between observed and calculated d[HONO]/dt.
Generally, the observed and calculated d[HONO]/dt agreed better with each other outside
the HONO explosive growth periods, showing overestimations when $S_A$ was high. For the fog
cases, no $S_A$ was available to account for $P_{het}$, however, for the haze case on 11$^{th}$ Nov (Fig. 6c) it
can be noted that by accounting for the photo-enhanced NO$_2$ conversion, an overestimation in $P_{het}$
occurred between 14 to 18 LT, while the rapid HONO formation in the morning could not be
explained. This further suggests that the observed discrepancies in HONO production have mainly
been caused by uncertainties in the heterogeneous formation (NO$_2$ uptake coefficient) estimates.
The fact that HONO drastically increased while NO$_2$ varied little (9:30 to 11:30, 5$^{th}$ Nov. and 6:30
to 8:30, 11$^{th}$ Nov.) or hardly increased even under drastic increases of NO$_2$ (8:30 to 11:30, 14$^{th}$
Nov.), but displayed explosive growth with increasing NH$_3$, could not be explained by currently
known HONO sources (direct emission or gas phase reactions). Additionally, these rapid
increasing HONO phenomena were all observed under foggy or severe haze with high RH
conditions, which further affirms the suspicion that the HONO increase was caused by
heterogeneous conversion of $NO_2$. Under such conditions, $S_A$ was not the controlling factor
determining the conversion of $NO_2$, so which mechanism could have been behind such rapid
HONO production?
**4.2 Heterogeneous HONO formation mechanism**
As manifested in Sect. 4.1, the unknown HONO source and the overestimates in HONO
production were both linked to our limited understanding on the heterogeneous HONO formation
mechanism. In this section, we try to evaluate the relative contribution of the currently known
heterogeneous HONO formation pathways (R1 and R2) and reveal the reason for their limitations
in explaining the observed HONO growth.
To evaluate which process (R1 or R2) was dominating the heterogeneous production of
HONO, we assume that HONO was produced in aerosol and fog water simultaneously via R1 and
R2. Since measurements of fog liquid water content or fog droplet surface area density were not
made, we cannot directly quantify the absolute HONO production in fog. However, we can make
a few assumptions to compare the relative HONO contribution via R1 and R2. First, it was
assumed that the observed sulfate production (d[SVI]/dt) was caused by the reaction of $SO_2$ with
$H_2O_2$, $O_3$, $NO_2$, transition metal ions (TMI: $Fe^{3+}$ and $Mn^{2+}$). Calculations were performed
according to Cheng et al. (2016), using the same pH dependent TMI concentrations and the
actually measured $SO_2$, $H_2O_2$, $O_3$ and $NO_2^*$ concentrations (Table S2). For the two fog episodes
on $4^{th}$ and $5^{th}$ Nov. 2016, the mean diameter of fog droplets was assumed to be 7.0 μm and the
liquid water content was assumed to be 0.3 g $m^{-3}$ according to Shen et al. (2018). For the haze
episodes on the $11^{th}$ and $14^{th}$ Nov. 2016, the mean aerosol diameter under ambient conditions was
estimated to be 0.65-1.22 and 0.9 μm (size-resolved volume contribution of aerosol particles in
dry state peaks near 500 nm), while the liquid water content was calculated to decrease from
$5.7×10^{-4}$ to $6.4×10^{-5}$ g $m^{-3}$ on the $11^{th}$ Nov and assumed to be 0.01 g $m^{-3}$ on the $14^{th}$ Nov. during
the transition from fog to haze. The sulfate production rate and relative contribution of each
oxidation pathway to the total sulfate production rate was obtained and depicted in Fig.7. For the
two fog episodes, assuming pH=6, the estimated average sulfate production rates are 16.6 and 49.1
μg $m^{-3}$ $h^{-1}$, respectively, approximately 3 and 7 times of that observed within $PM_{2.5}$, which might
be an underestimation, considering the liquid water content of fog droplets are at least a magnitude
higher than that of aerosols. For the two haze episodes, using the pH values (3.8-5.19 and 4.15 for
the two haze events, respectively) estimated using ISORROPIA (forward mode and metastable
assumption (Song et al., 2018)), the estimated average sulfate production rates are 0.33 and
0.94 $\mu g\ m^{-3}\ h^{-1}$, about 38% and 20% of that observed within $PM_{2.5}$. Following the calculations of
Cheng et al. (2016), we have considered the influence of ionic strength on the reaction rates and
set constraints on the maximum ionic strength ($I_{max}$), which might have caused underestimations
for all reaction routes, since the calculated ionic strength commonly exceeded $I_{max}$. Underestimated
transition metal ion concentrations may also be partly responsible for the underpredicted sulfate
production, since the TMI catalysis route has recently be pointed out to be the dominant $SO_2$
heterogeneous oxidation pathway (Shao et al., 2019) under low pH conditions. Additionally, there
also might be other neglected $SO_2$ oxidation pathways, which will lead to overestimates in the
sulfate fraction produced by the $NO_2$ oxidation pathway. Therefore, we can only yield an upper
limit for the HONO production rate of R1:
$$\frac{d[HONO]}{dt}_{R1} = 2 \times frac_{SO_2+NO_2} \times \frac{d[SVI]}{dt}_{obs},$$   (Eq.7)
where $frac_{SO_2+NO_2}$ is the contribution fraction of the $NO_2$ oxidation pathway to the total sulfate
production. Note that the calculated HONO production rate can only represent the production
within $PM_{2.5}$.
By further assuming that all the observed $HNO_3$ and nitrate production ($d[HNO_3+NO_3^-]/dt$) was
caused by reaction R2 and by the reaction of $NO_2$ with OH radicals ($k_{NO_2+OH}=3.2\times10^{-12}\ cm^3\ s^{-1}$),
the HONO production rate of R2 would be:
$$\frac{d[HONO]}{dt}_{R2} = \frac{d[HNO_3+NO_3^-]}{dt}_{obs} - k_{NO_2+OH}[NO_2^*][OH].$$   (Eq.8)
The contribution fraction of the two reactions to the heterogeneous HONO production in aerosol
and fog liquid water content can be calculated by:
$$f_{R1} = \frac{d[HONO]}{dt}_{R1} / \frac{d[HONO]}{dt}_{R1+R2} \quad \text{and}$$   (Eq.9-1)
$$f_{R2} = \frac{d[HONO]}{dt}_{R2} / \frac{d[HONO]}{dt}_{R1+R2}.$$   (Eq.9-2)

Assuming the pH of fog droplets falls within the range of 4 to 6, $f_{R2}$ was estimated to range

from range from 82.2 to 99.7% and from 86.8 to 99.8% during the 4[th] and 5[th] Nov. 2016,
respectively. For the two haze events on 11[th] and 14[th] Nov., the $f_{R2}$ corresponding to the pH values
modelled by ISORROPIA would be 99.7% and 98.0%.

These results suggest that, reaction R2 is the dominant contributor to the heterogeneous

HONO production, while R1 is more important under high pH conditions. Under the assumed
upper limit of pH, R1 could have contribute up to 17.8% and 13.2% to the observed HONO growth
during the two fog events. This is in accordance with results from Wang et al. (2016) and Cheng
et al. (2016), which suggested that R1 was more likely to happen during fog episodes or under
$NH_3$ neutralized conditions (3,4). For the two haze events, R1 contributed very little (0.3% and
2%) to the observed HONO growth.

Since R2 seems to be the dominant contributor to the observed HONO production, it is

important to evaluate whether the parameterizations in current literature can accurately describe
the HONO production process of R2. The HONO production rate of R2 is typically parameterized
as in Eq.4, where the $NO_2$ reactive uptake coefficient, $NO_2$ concentration and the surface area
density of fog droplets/aerosol particles are the controlling factors of the $NO_2$ uptake, as opposed
to the pH of the water droplets (Li et al., 2018b;Yabushita et al., 2009). Based on the $NO_2$ reactive
uptake coefficient ($\gamma_{NO_2}$) range of $1\times10^{-4}$ to $1\times10^{-3}$ in Yabushita et al. (2009) and Li et al. (2018b),
which is represents the upper limit for currently reported $\gamma_{NO_2}$, we have calculated the HONO
production rate of R2 under different conditions (Fig.S5). During foggy conditions, the HONO
production rate would be higher than 1 ppb (ppb $NO_2\cdot$h)$^{-1}$. $NO_2^*$ during the two fog episodes
ranged between 37 to 40 ppb, therefore, the HONO production rate would have been higher than
40 ppb h$^{-1}$. However, no rapid increase of HONO was observed unless $NH_3$ was simultaneously
increasing. The same conclusion can be reached for hazy conditions. If we had used a constant
$\gamma_{NO_2}$ of $1\times10^{-4}$ for hazy conditions in the budget analysis, the calculated $P_{het}$ would significantly
overestimate the HONO production when relative humidity was high and large ambient $S_a$ were
observed, while it would fail to reproduce the growth in HONO on the morning of the 11[th] Nov.
2016 (Fig. S6). The $\gamma_{NO_2}$ parameterization in Sect. 4.1, which accounted for photo-enhancement,
also failed to explain the morning growth of HONO and resulted in overestimated HONO
production during the afternoon. These results indicate that $\gamma_{NO_2}$ is not a constant, the currently
proposed $\gamma_{NO_2}$ parameterization schemes for R2 are missing the important impact of $NH_3$. The
$\gamma_{NO_2}$ range used in Yabushita et al. (2009) and Li et al. (2018b) highly overestimated HONO
production, when $NH_3$ was not abundant enough, while it was insufficient to explain the observed
HONO production with the growth of $NH_3$.

Recent theoretical simulation results ascertain that $NH_3$ can promote the hydrolysis of $NO_2$

and contribute to HONO formation via R2 by reducing the free energy barrier of the reaction and
stabilizing the product state (Li et al., 2018a). This conclusion is consistent with the observed
phenomena that HONO only increased rapidly when $NH_3$ was simultaneously increasing.
Considering the influence of $NH_3$ and sulfate on the aerosol pH, under our observed $NH_3$
concentration range, $NH_3$ has negligible impact on pH values (Guo et al., 2017), especially under
high RH conditions. This further proves that the $NH_3$-promoted hydrolysis of $NO_2$ is independent
of the pH value. Another phenomenon worth noting is that, in Case 3, HONO was increasing
rapidly even under the drastic decrease in ambient RH, which demonstrates that the impact of $NH_3$
on HONO formation should be even more important than that of aerosol liquid water content.
However, the hydrolysis of $NO_2$ needs water to be involved, thus, the importance of water content
under different conditions remains to be elucidated.

To further investigate the acceleration effect of $NH_3$ on the hydrolysis of $NO_2$, we have

examined the correlations between the $NO_2$*-to-HONO (HONO/ $NO_2$* ratio), $NO_2$*-to-$NO_3^-$
($NO_3^-$/ $NO_2$* ratio) conversion efficiencies and the $NH_3$ concentration during the entire field
campaign (Fig.7). Note that only data points during nighttime (18 pm to 6 am) and with ambient
RH higher than 80% are displayed in Fig.8. Daytime data were excluded, because HONO would
quickly photolyze as soon as sunlight was available. Even if there was rapid HONO production,
the corresponding increase of HONO might not be observable due to its quick photolysis. The
reason for only including data with ambient RH higher than 80% is that the quick hydrolysis of
$NO_2$ requires water to be involved. However, the overall hygroscopicity of ambient aerosols during
this field campaign was relatively low, with an average hygroscopicity parameter $\kappa$ of 0.14, and
the volume contribution of liquid water to the total volume concentrations of ambient aerosols was
quite low when ambient RH is below 80% (Kuang et al., 2018). The correlation coefficient between
HONO/$NO_2^*$ ratio and the $NH_3$ concentration reaches 0.68, while that between $NO_3^-$/$NO_2$ ratio
and $NH_3$ concentration only reaches 0.53, since the source of $NO_3^-$ is much more complicated than
that of HONO. These results have further verified that $NH_3$ promotes the $NO_2$ hydrolysis and
HONO production. The correlation of HONO/$NO_2^*$ to $NH_3$ is highly nonlinear, HONO/$NO_2^*$
increases rapidly with $NH_3$ when $NH_3$ reaches above 10 ppb. In retrospect to Sect. 3 and Fig. 3, it
can be concluded that, under relatively cleaner conditions, the heterogeneous HONO formation
was mainly limited by particle surface area, while under polluted conditions, $NH_3$ concentration
was the dominant limiting factor.

**4.3 Feedback between HONO formation and inorganic secondary aerosol formation**

According to the discussions in Sect.4.2, $NH_3$ promotes the hydrolysis of $NO_2$, producing
nitrate and most of the observed HONO. However, the connection between the $NH_3$ promoted
hydrolysis and the simultaneous rapid sulfate production remains unexplained. As was already
discussed in Sect.4.2, the sulfate production rate calculated based on currently known $SO_2$
oxidation pathways largely underestimates the observed sulfate growth, indicating that there might
be neglected oxidation pathways. Li et al. (2018b) pointed out that $NO_2$ can oxidize S(IV)
indirectly via free radical mechanism (the involved reactions RS1 to RS5 proposed in Li et al.
(2018b) listed in the supplement). The key step of the proposed S(IV) oxidation pathway is the
photolysis of HONO to produce OH radicals (RS1). OH can oxidize S(IV) to form bisulfate or
sulfate through reaction RS2 and produce $HO_2$. $HO_2$ can react with NO to produce $NO_2$, or react
with itself to produce $H_2O_2$. As was depicted in Fig.6, the radiation during the fog/haze events was
already strong enough to photolyze the produced HONO and release OH radicals at the same rates
as $L_{pho}$ in Sect. 4.1, indicating there was strong OH production, especially near the end of the
events. For the two fog events, no AOD measurements were available. Assuming AOD=2.5 for
foggy conditions, the lifetime of HONO (only considering the photolysis process) were estimated
to decrease from 4.2 to 1.1 h, 1.7 to 1.1 h during the growth of HONO and to drop to 1.0 and 0.7 h
by the time of the drastic decreases in HONO. In the haze event on the 11[th] Nov., AOD
measurements were also not available due to cloud coverage, however, sensitivity study shows
that the calculated HONO lifetime are much more sensitive to the AOD as opposed to the COD
values (increasing 3.1 and 0.4 h per 0.1 increase in AOD and COD, Fig.S7). The HONO lifetime
dropped from 2.0 h (by the time of the HONO peak) to 1.1 h (by time of the HONO decrease).
During the case on the 14[th] Nov. 2016, the relative humidity decreased from 100% (10:00-11:00)
to 86% (11:30), suggesting that this was a fog dissipation process. The HONO lifetime was
estimated to be 1.7 h between 10:00 to 11:00, proving that the photolysis process was relatively
weaker during the rapid increase of HONO. The estimated HONO lifetime rapidly decreased to
0.6 h by 12:00, resulting in accelerated HONO dissociation and OH production. The increase in
$H_2O_2$ observed during and after the increase of HONO, might be an indirect evidence of the $HO_2$
production and occurrence of RS2. The observed $H_2O_2$ concentrations were much higher than the
assumptions of 0.01 ppb made in Cheng et al. (2016), which was also pointed out by Ye et al.
(2018). Under the assumed pH range for fog and the calculated pH range for aerosol, the estimated
sulfate production was dominated by the $SO_2$ oxidation via $H_2O_2$ (Fig.7). This indicates that both
the calculated and the yet unexplained sulfate production were linked to the photolysis of HONO.

$NH_3$ promoted the hydrolysis of $NO_2$, producing HONO and nitrate. HONO easily photolyzes

releasing OH radicals, which further converted to $HO_2$ and $H_2O_2$. The highly oxidative free radicals
and $H_2O_2$ collaboratively boosted the formation of sulfate. Hence, diurnal variations of $NH_3$ should
have exerted significant influences on the diurnal variations of HONO and inorganic aerosol
chemical components (sulfate, nitrate and ammonium, SNA). The average diurnal variations of
$NO_2^*$, $NH_3$, HONO as well as $SO_2$ concentrations during this field campaign are shown in Fig.9a.
The average HONO concentration during nighttime is higher than that during daytime due to the
quick photolysis of HONO upon solar irradiation. The $NH_3$ concentration begins to increase in the
morning (near 8:00 LT) the reaches a plateau in the afternoon (8.5 to 15.5 ppb in average), and the
$SO_2$ concentrations shows a similar diurnal variation to that of $NH_3$. This type of diurnal variation
of $SO_2$ was also found by Xu et al. (2014), however, the cause of the common diurnal pattern
between $NH_3$ and $SO_2$ during this field campaign requires further investigation. The $NO_2^*$
concentration increases quickly in the afternoon and decreases in the evening.

As shown in Fig.9b, the increase of $NH_3$ from morning to the afternoon was accompanied

with the increase of mass fractions of nitrate and sulfate in $PM_{2.5}$ (The mass fractions of different
aerosol chemical compositions were obtained by using the measured dry state PNSD to calculate
volume concentration of $PM_{2.5}$, assuming that the density of aerosols in dry state is 1.5 g $cm^{-3}$ (Yin
et al., 2015). The results shown in Fig.9b indicate that the molecular concentration increase in
nitrate from the morning to the afternoon is much faster than that of sulfate, again supporting the
fact that the $NH_3$-promoted $NO_2$ hydrolysis, which only produces HONO and nitrate directly, was
the main contributor to the observed explosive HONO formation. The evident morning increase
of inorganic aerosol component fractions resulted in prominent increases of aerosol hygroscopicity,
displaying an average $\kappa$ anomaly of +0.04 during noontime (Fig.9c). From the morning to the
afternoon, the ambient RH decreases quickly, however, the increase of aerosol hygroscopicity can
retard the decrease of aerosol liquid water content and surface area density of ambient aerosols.
This might act as a positive feedback, further enhancing the hydrolysis of $NO_2$ as well as the nitrate
and sulfate formation.
**5. Summary and atmospheric implications**
Explosive HONO growth (observed maximum d[HONO]/dt=16.1 ppb h$^{-1}$) was observed
for the first time on the NCP during fog and haze episodes with high RH conditions, only occurring
with evident increases in $NH_3$, indicating that $NH_3$ is the key factor promoting the hydrolysis of
$NO_2$, resulting in rapid HONO and nitrate formation. $NH_3$ concentrations during the observation
period exhibit a distinct diurnal variation with an increase in the morning and a peak in the
afternoon (8.5 to 15.5 ppb in average). The increase of $NH_3$ promotes the hydrolysis of $NO_2$, giving
significant rise to HONO and nitrate concentrations. Produced HONO released OH radicals upon
photolysis, which further oxidized $SO_2$ to sulfate through gas phase and heterogeneous reactions.
Therefore, the significant growth of $NH_3$ in the morning determined the increase in nitrate, sulfate
and ammonium as well as that of aerosol hygroscopicity, which as a positive feedback retards the
decrease in atmospheric liquid water content and further enhances the hydrolysis of $NO_2$ as well
as the nitrate and sulfate formation.
Results in this paper reveals that the $NH_3$-promoted $NO_2$ hydrolysis is a significant source
of HONO, especially under polluted conditions, which provides direct insight into the missing
daytime source of HONO on the NCP. Results in this paper also shed light on the recent
controversy of how $SO_2$, pH and $NH_3$ are involved in heterogeneous HONO production. It was
clarified that in the HONO production, $SO_2$ took a minor part during fog events and an insignificant
part during haze events, the observed growth in sulfate was dominantly the byproduct of the
HONO photolysis, confirming again the importance HONO as an OH source and its crucial role
in atmospheric chemistry.
These results have demonstrated the critical role and contribution of $NH_3$ in the formation
of photochemical and aerosol pollution on the North China Plain. Effective control measures are
urgently called for to reduce $NH_3$ emissions, which would simultaneously benefit the
photochemical and aerosol pollution abatement through the reduction of HONO production.
**Author contribution**
WX designed the experiment and YK led the research. YK, JT, GZ, YB, YY, CS and LL
were responsible for the aerosol measurements in the experiment, WY helped with the IGAC

measurements. WX made the trace gas measurements with the help of ZG, WL and XX. YK and WX analyzed the data and wrote the paper.

**Acknowledgments, Samples, and Data**

This work is supported by the National Key R&D Program of China (2016YFC0202300), the National research program for key issues in air pollution control (DQGG0103) and the National Natural Science Foundation of China (41505107 and 41590872). We thank Wei Peng from Beijing Met High-Tech Co., Ltd. for his help with the maintenance of the IGAC instrument.

**Data availability**. The data used in this study are available from the corresponding author upon request (kuangye@jnu.edu.cn)

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

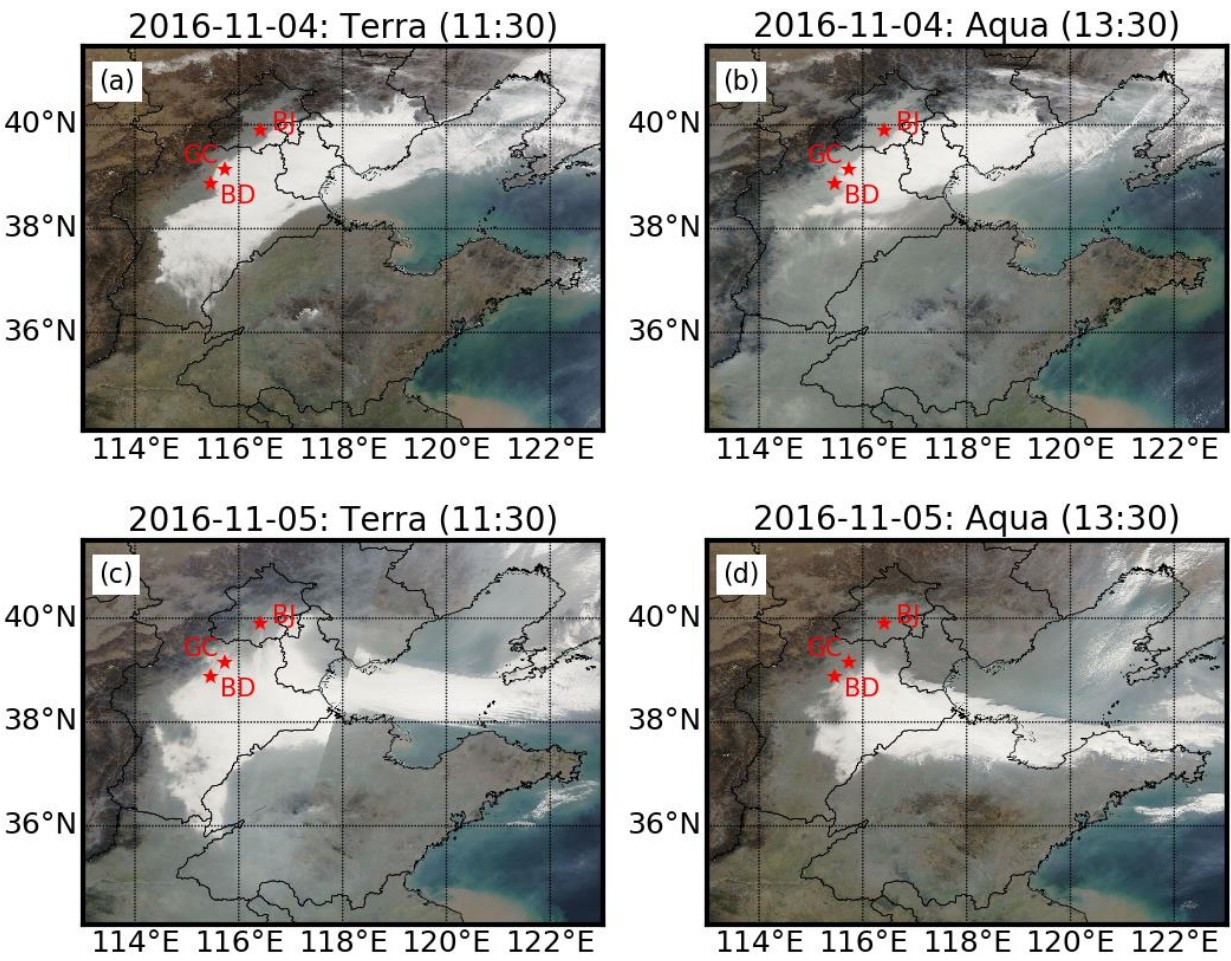

**Figure 1**. MODIS Terra (a,c) and Aqua (b,d) satellite images in 04[th] Nov. (a,b) and 5[th] Nov. 2016
(c,d), star markers are locations of Gucheng (GC: the observation site), Baoding (BD) and Beijing
(BJ).

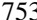

**Figure 2**. Time series of ambient **a)** RH; **b)** HONO; **c)** sulfate, nitrate, ammonium; **d)** $NH_3$, $NO_2^*$ and $SO_2$ during the observation period.

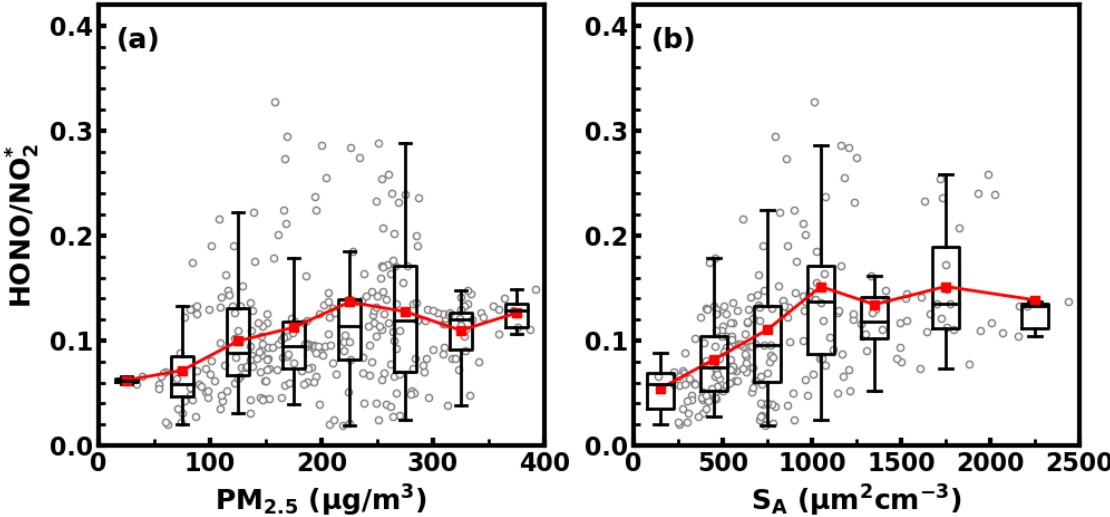

Figure 3 Boxplots displaying the variation of HONO/NO$_2^*$ with a) PM$_{2.5}$ concentration and b) ambient aerosol surface area density.


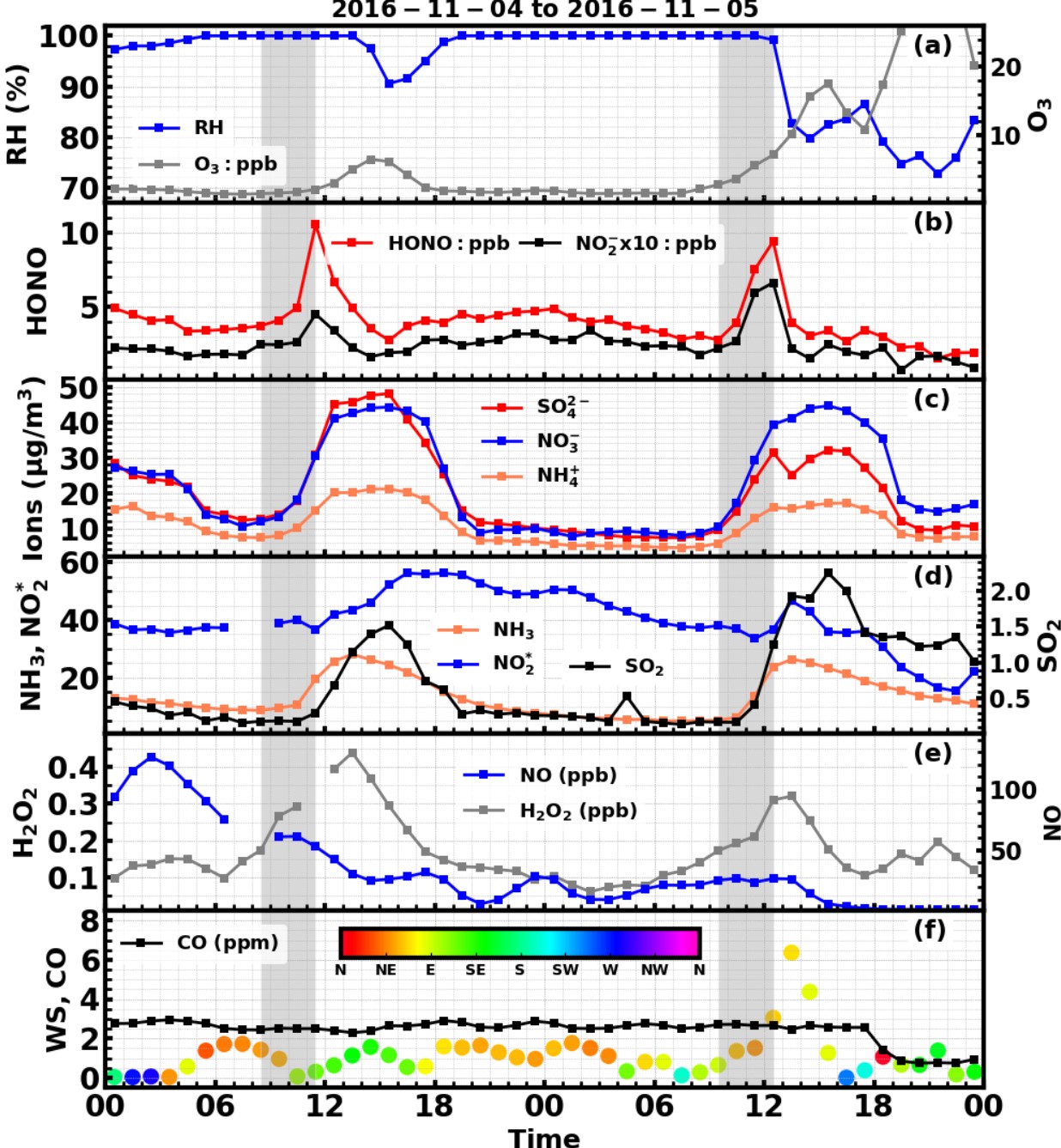

**Figure 4** Time series of ambient **a)** RH,$O_3$, **b)**HONO, $NO_2^-$, **c)** $SO_4^{2-}$, $NO_3^-$, $NH_4^+$, **d)** $NH_3$, $NO_2^*$,
$SO_2$, **e)** NO, $H_2O_2$, **f)** CO, wind speed and wind direction (colors of scatter points ) from 4[th] to 5[th]
Nov. 2016. Gray shaded areas represent periods of rapid increase of HONO.

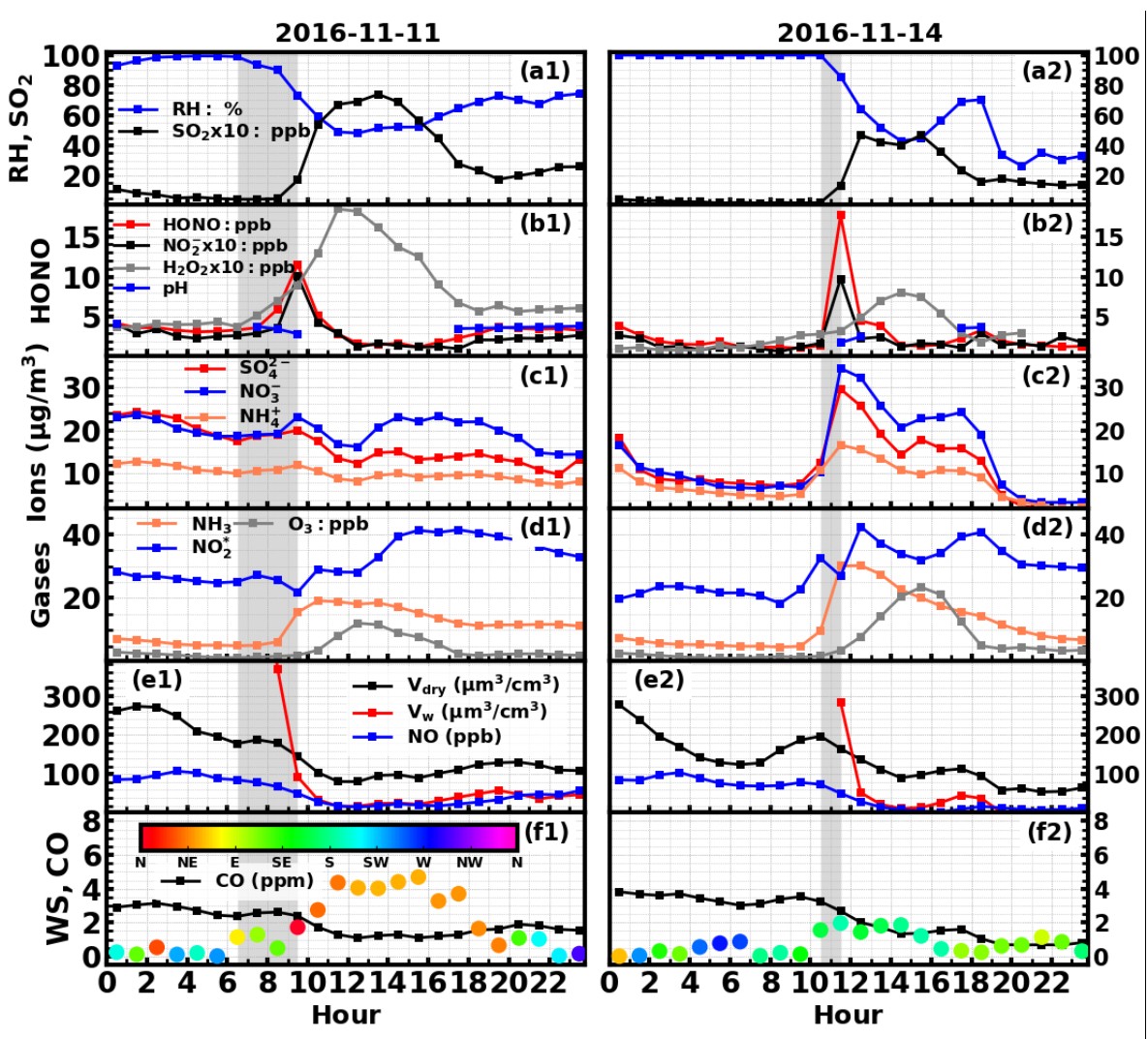

**Figure 5** Time series of ambient **a)** RH, SO$_2$, **b)** HONO, NO$_2^-$, H$_2$O$_2$, aerosol pH, **c)** SO$_4^{2-}$, NO$_3^-$,
NH$_4^+$, **d)** NH$_3$, NO$_2^*$, O$_3$, **e)** NO, volume concentrations of PM$_{2.5}$ in dry state (V$_{dry}$), volume
concentrations of liquid water (V$_w$), **f)** CO, wind speed and wind direction during **1)** 11$^{th}$ Nov. 2016
and **2)** 14$^{th}$ Nov. 2016. Gray shaded areas represents periods of rapid increase of HONO.
772

773

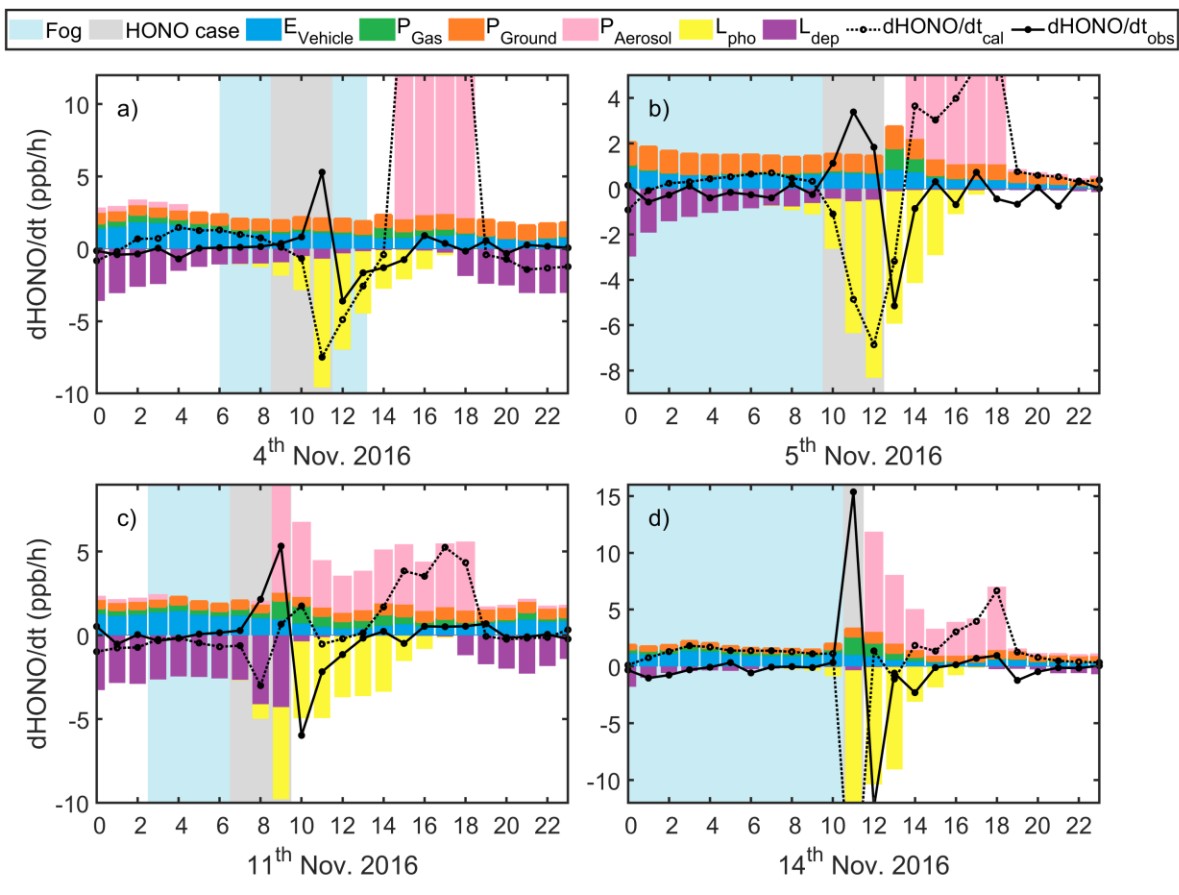

**Figure 6** Estimated HONO emission from vehicles (blue), gas phase production (green), production on ground (orange) and aerosol surface (pink), loss through photolysis (yellow) and dry deposition (purple), as well as the calculated (dotted black) and actually observed (solid black) d[HONO]/dt on a) 4th, b) 5th, c) 11th and d) 14th Nov. 2016.


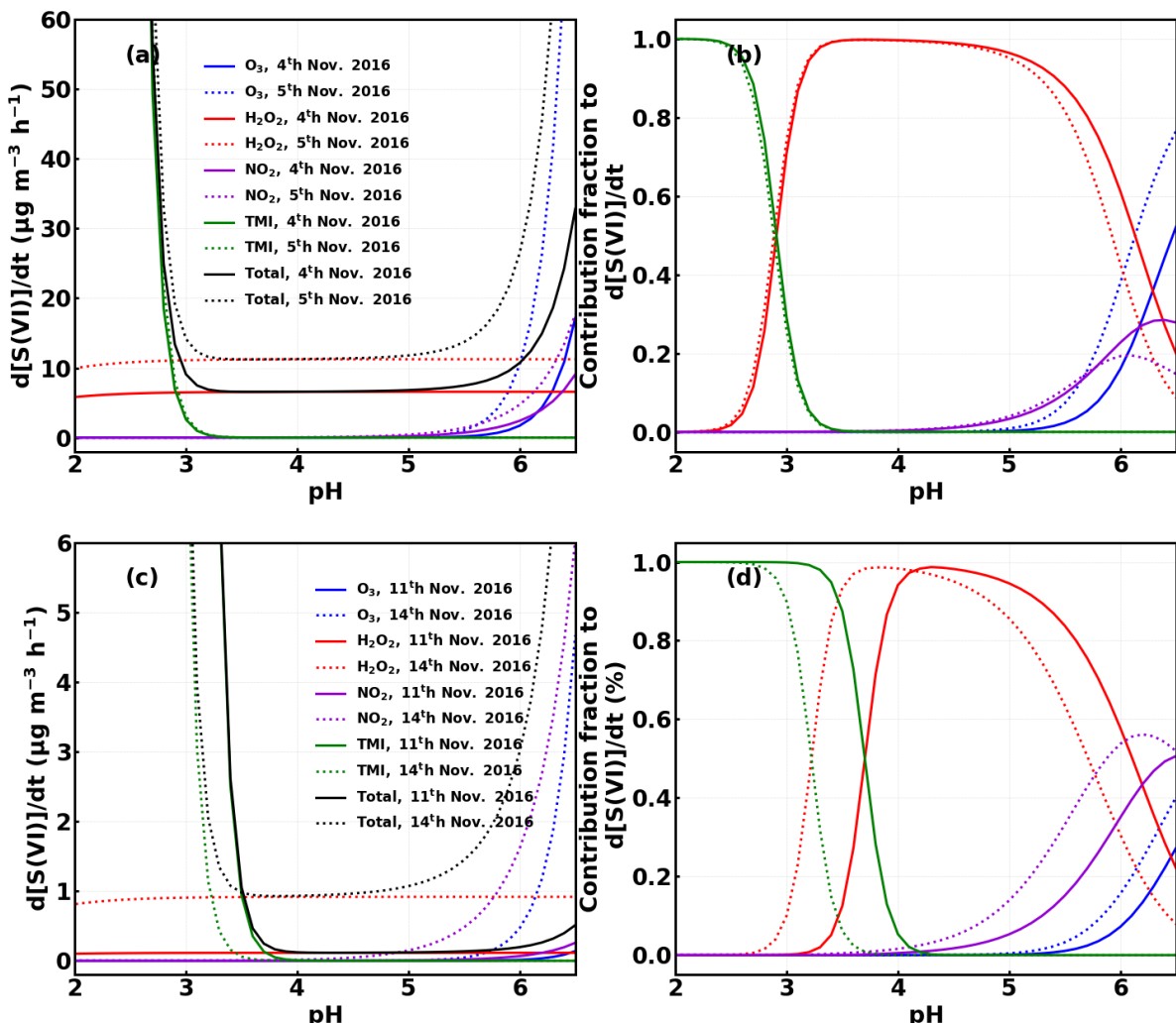

**Figure 7** Calculated average sulfate production (a,c) and contribution fraction b,d) from $SO_2$
oxidation by $H_2O_2$, $NO_2^*$, $O_3$, TMI under different pH values using methods described in (Cheng
et al., 2016) for the case episodes on 4th, 5th, 11th and 14th Nov. 2016.

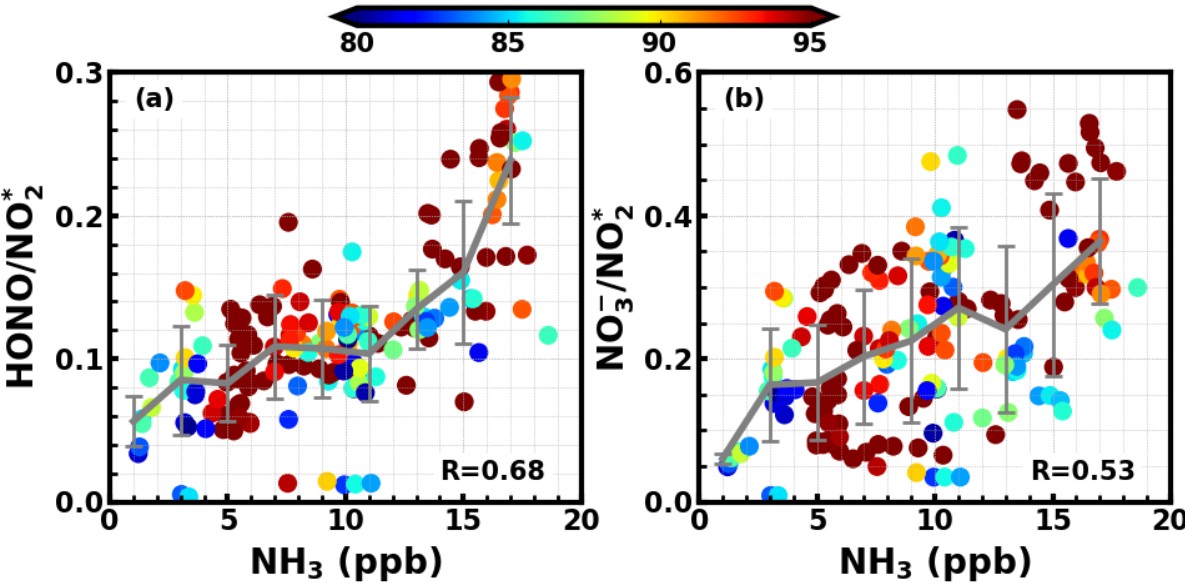

**Figure 8** The relationship between $NH_3$ concentration and **a)** HONO/ $NO_2^*$ ratio; **b)**
nitrate/nitrogen dioxide ratio ($NO_3^-/NO_2^*$); Colors of scatter points represent ambient RHs and the
color bar is shown on the top.


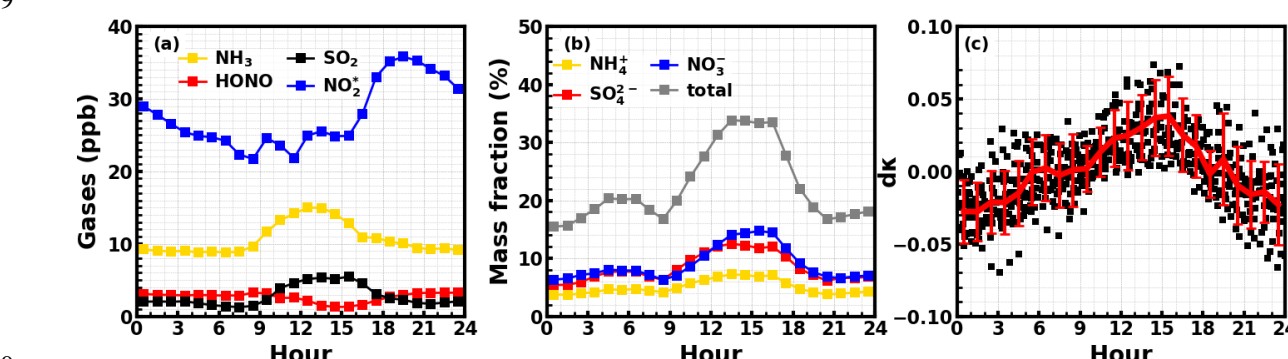


**Figure 9.** **(a)** Average diurnal variations of Gases; **(b)** Average diurnal variations of mass
fractions **of** nitrate, sulfate and ammonium; **(c)** Diurnal variations of aerosol hygroscopicity, dκ is
the anomaly to the daily mean κ.