# Peer review of "NH3-promoted hydrolysis of NO2 induces explosive growth in HONO"

_Atmospheric Chemistry and Physics, 2018_

## Referee Comment (RC1) · Anonymous Referee #1 · 4 Jan 2019

The article "NH3-promoted hydrolysis of NO2 induces explosive growth in HONO" discussed the mechanism behind explosive HONO formation during field observation in a rural site in North China. In general, the phenomenon, confidence of related evidence were sufficient to show the role of NH3 in HONO productions via heterogeneous reaction during fog/smoke events. The observation data was well linked to possible atmosphere processes, which might greatly promote the understanding of HONO sources and thus providing new insights of pollution control strategies for China. Yet, the authors should address several minor points to make the narrative as well as the deduction more convincible. Minor suggestions First, it is well known that the HONO is extremely reactive especially during daytime. For most of the cases, the author noted that the rarely seen Ozone was the evidence that there was no sufficient sunlight. It

seems that the Ozone concentration was used as an indicator of UV radiation and possible photochemistry reactions (line213-line218). But the Ozone could be titrated by NO, which was often measured a high level during nighttime in North China. Therefore, even the Ozone was observed to be nearly zero, there might be enough UV radiation for the quick HONO photolysis. This leads to a further question - can we trust the HONO measurements by a denuder system? The annular denuder method of detecting might have artefacts regarding to measuring HONO due to: 1. Hydrolysis of NO2 onto wet surface; 2.Aqueous reaction of S(IV) with NO2 in the solution(Spindler, Hesper et al. 2003, Nie, Ding et al. 2015). The second reaction could likely be accelerated in the presence of ammonia as reported in the previous studies (Cheng, Zheng et al. 2016, Wang, Zhang et al. 2016). Therefore, it is strongly recommended that the authors should conducted some validation of the HONO data from IGAC, given the fact that the HONO data obtained by denuder system need further calculation/reprocessing.

The second point will be lying in the mechanism discussions. Though R1 could not explain the increasing nitrate (line 260-261), but attributing all the SIA (secondary inorganic aerosols) increase due to HONO formation and thus denying the role of R1 seems to be assertive. Actually, the nitrite in the aqueous phase might have produced OH radicals in aerosol liquid water or fog droplets (Vione, Maurino et al. 2006). It would be good to illustrate or maybe quantify the relative contribution of R1 v.s. R2 to HONO production as well as SIA production.

Besides the above comments, some technical notes shall be taken as well (mostly on the writing and texts): 1. Line110-111: "Under highly polluted conditions such as our site". Might have wrong grammar used. 2. Figure 3. The time label on X axis causes misunderstanding, might change to Date-Time format. 3. Figure 4. The unit of aerosol composition (nitrate/sulfate/ammonium) should be in mass concentration.

References: Cheng, Y., G. Zheng, C. Wei, Q. Mu, B. Zheng, Z. Wang, M. Gao, Q. Zhang, K. He, G. Carmichael, U. Pöschl and H. Su (2016). "Reactive nitrogen chemistry in aerosol water as a source of sulfate during haze events in China." Science

Advances 2(12): e1601530. Nie, W., A. J. Ding, Y. N. Xie, Z. Xu, H. Mao, V. M. Kerminen, L. F. Zheng, X. M. Qi, X. Huang, X. Q. Yang, J. N. Sun, E. Herrmann, T. Petäjä, M. Kulmala and C. B. Fu (2015). "Influence of biomass burning plumes on HONO chemistry in eastern China." Atmos. Chem. Phys. 15(3): 1147-1159. Spindler, G., J. Hesper, E. Brüggemann, R. Dubois, T. Müller and H. Herrmann (2003). "Wet annular denuder measurements of nitrous acid: laboratory study of the artefact reaction of NO2 with S(IV) in aqueous solution and comparison with field measurements." Atmospheric Environment 37(19): 2643-2662. Vione, D., V. Maurino, C. Minero, E. Pelizzetti, M. A. J. Harrison, R.-I. Olariu and C. Arsene (2006). "Photochemical reactions in the tropospheric aqueous phase and on particulate matter." Chemical Society Reviews 35(5): 441-453. Wang, G. H., R. Y. Zhang, M. E. Gomez, L. X. Yang, M. L. Zamora, M. Hu, Y. Lin, J. F. Peng, S. Guo, J. J. Meng, J. J. Li, C. L. Cheng, T. F. Hu, Y. Q. Ren, Y. S. Wang, J. Gao, J. J. Cao, Z. S. An, W. J. Zhou, G. H. Li, J. Y. Wang, P. F. Tian, W. Marrero-Ortiz, J. Secrest, Z. F. Du, J. Zheng, D. J. Shang, L. M. Zeng, M. Shao, W. G. Wang, Y. Huang, Y. Wang, Y. J. Zhu, Y. X. Li, J. X. Hu, B. Pan, L. Cai, Y. T. Cheng, Y. M. Ji, F. Zhang, D. Rosenfeld, P. S. Liss, R. A. Duce, C. E. Kolb and M. J. Molina (2016). "Persistent sulfate formation from London Fog to Chinese haze." Proceedings of the National Academy of Sciences of the United States of America 113(48): 13630-13635.

---

## Referee Comment (RC2) · Anonymous Referee #2 · 21 Jan 2019

The paper "NH3-promoted hydrolysis of NO2 induces explosive growth in HONO" by Wanyun Xu et al is joining the long series of scientific work aiming at elucidating the HONO sources that have been published these past twenty years. Considering the major role of HONO in the initiation of the photo-oxidation cycles in the troposphere, any significant work related to the processes that give birth to this key molecule are necessarily important (Kleffmann et al, 2007).

Wanyun Xu et al are mostly founding the exploitation of their results on a methodology based on recent papers only and are often disregarding the precious findings of the early times. Nevertheless, they provide here an attempt to exploits a limited set of data obtained in heavily polluted environment that is not without any merit.

Overall, I have many minor points to discuss but, for me, one major point is shining

a doubtful light over the whole study: it concerns the reliability of the HONO/NO2-measurements themselves. The reliable measurement of HONO at low level in the atmosphere has been an analytical challenge for decades. Many groups have worked on various analytical concepts ranging from long path spectroscopy, optical cavities, ionic chromatography or dye formation combined with absorption within waveguide tubing or HPLC analysis...

Because of HONO high reactivity the risk of underestimation of its concentration is often high. In the same time, because of the multiplicity of its heterogeneous sources the risk of positive artefact and unwanted HONO generation in/nearby the system is high too. This is why, even when the measurement principle itself was mature, the sampling condition was often found to be a key parameter for trustable measurements which has led to important work on the design of inlets, minimizing surfaces, choosing material...

Each of these instrumental concept has required extensive characterization works and a few inter-comparison exercises have demonstrated how large the discrepancies can be (Keuken et al, 1990; Stutz et al, 2009; Kleffmann et al, 2006; Pinto et al, 2014).

In the present paper, the whole experimental strategy relies on the performance of the so-called In situ Gas and Aerosol Compositions Monitor (IGAC, Fortelice International Co.,Taiwan) as both HONO/NO2- and NH3 concentrations – the two key species of the present study - are monitored using this instrument. The available information about this device are scarce: IGAC consists in a combination of a wet annular denuder and a particle into liquid sampler. Unfortunately it has been poorly characterized in general and none of the reference provided in the paper are relevant for HONO measurements. In particular, while citing Liu et al, 2017a to claim "the instrument has shown good performance in the past" or quoting Young et al, 2016, one can only regret that nothing in these papers really concern nitrous acid or nitrite ions measurements. Further, Young et al, 2016 indicate that the performance of IGAC were poor concerning the measurement of ammonia.

On my side, considering the IGAC experimental device and condition of use, I especially worry about the use of "a dilute H2O2 solution to collect the gases". If one refer to Young et al, 2016 the "dilute solution" is a 5x10-3 M solution (why not mentioning the concentration in the experimental section?) which is used to "assure the oxidation of SO2 to SO4– and prevents microbial growth ". For me, it is highly probable that such a concentration of such a strong oxidation agent could induced artefacts in the HONO measurement: - in the absence of precursors, it may induce a negative artefact by oxidizing nitrites to nitrates but, on the contrary - in the presence of enough reduced nitrogenous species (such as ammonia) it may forms HONO. In this case this would both affect NH3 and HONO measurements and would probably lead to a correlation between both species (if ammonia is in excess).

Considering the poor level of details provided in the experimental section, the lack characterization experiments demonstrating the ability of IGAC to measure HONO (especially in the presence of ammonia) and the strong suspicion of artefacts exactly relevant from the main paper conclusion, I strongly recommend to provide the experimental evidences that demonstrate the suitability of the measurement protocol for both NH3 and HONO before considering any publication.

Other Major issues

Line 213-214: "The O3 concentration stayed near zero, which means that UV radiation was weak." This statement is clearly wrong. From the few NO data that the author disclose to the reader one can see that NO values are typically ranging from 20 to 100 ppb. With such high values, no wonder why O3 remain low: it is clearly titrated by NO. One can understand that the lack of spectral radiometer measurements is an issue (see later) but O3 data in a polluted environment can certainly not be used as a proxy for UV radiation strength.

Line 226: Equation 1 is strongly oversimplified.

On the HONO sinks side one clearly miss - photolysis that can certainly not be neglected. Even if radiation measurements are not available, the authors manage, later on in the paper, to evaluate some values that could be used here. Another approach would be to provide an upper limit evaluating the J value above haze using TUV for example (see Madronich et al, 1988 and Tie et al, 2003) - deposition can be taken into account by using as deposition velocity the value given by Stutz et al., 2002, for example. In addition in the presence of hydrometeors, one clearly miss the loss processes onto/into haze droplets. On the HONO source side, may well identified processes are missing such as direct emission and heterogeneous HONO formation from conversion of NO2 on ground surface and aerosol surfaces.

Line 304-306 then line 326-329: In these section the photolysis of HONO is described being rapid (which is probably true) while it has been neglected earlier. I think the manuscript need reorganization to discuss more coherently the photochemistry of HONO under these conditions.

Minor issues

Figure 2: it is somewhat disturbing that the figure does not displayed all the data acquired. In particular (but not only) the absence of NO and ozone data is clearly a problem. Furthermore, the use of "ppb" for aerosol composition is confusing: is it related to the whole volume of air? Is it related to the whole aerosol quantity. Please use more straightforward units here.

Line 98 – 108: The experimental description of the instrument, the inlet and the protocol is insufficient.

Line 120: The authors indicate the use of a NOx monitor 42CTL from thermo. It is not clear if this instrument is equipped with a Mo-based converter, a Blue light converter or both. In any case, the risk of interferences with HONO on the NO2 and NOx channels are high (through the conversion of HONO on heated Mo – see Dunlea et al, 2007 for example - or through its photolysis by the blue light. During some part of this field campaign the HONO values can be as high as 20 % of NO2. In this case it would be

necessary to evaluate the cross-sensitivities of NO2 and HONO in the configuration of the chemiluminescence monitor used.

Line 127: "wavelength" is misspelled

Line 229: The value of 10ˆ6 radicals/cm3 is taken as "typical for noontime haze condition" and later used in the equation 1. Even if the order of magnitude of this guess is probably not too wrong there is no reference provided. Furthermore, I don't think that the scientific community have the necessary background to raise a "typical value" for these quite peculiar conditions. I would rather recommend to refer to published work such as Whalley et al, 2015 (field) or Tie et al, 2003 (large scale modeling)

Line 267-269: This statement is quite vague. Which anions are the authors referring to ? More explanation are needed.

References

Keuken, M. P., R. P. Otjes and J. Slanina (1990). Simultaneously Sampling of NH3, HNO3, HNO2, HCl. SO2 And H2O2 in Ambient Air by A Wet Annular Denuder System. Physico-Chemical Behaviour of Atmospheric Pollutants: Air Pollution Research Reports. G. Restelli and G. Angeletti. Dordrecht, Springer Netherlands: 92-97.

Kleffmann, J. (2007), Daytime Sources of Nitrous Acid (HONO) in the Atmospheric Boundary Layer. ChemPhysChem, 8: 1137-1144. doi: 10.1002/cphc.200700016

Stutz J., Hoon-Ju Oh, Sallie I. Whitlow, Casey Anderson, Jack E. Dibb, James H. Flynn, B. Rappengluck, B. Lefer, Simultaneous DOAS and mist-chamber IC measurements of HONO in Houston, TX, Atmospheric Environment, Volume 44, Issue 33, 2010, Pages 4090-4098, ISSN 1352-2310, doi: 10.1016/j.atmosenv.2009.02.003.

Kleffmann J., J.C. Lerzer, P. Wiesen, C. Kern, S. Trick, R. Volkamer, M. Rodenas, K. Wirtz, Intercomparison of the DOAS and LOPAP techniques for the detection of nitrous acid (HONO), Atmospheric Environment, Volume 40, Issue 20, 2006, Pages 3640-3652, ISSN 1352-2310, doi: 10.1016/j.atmosenv.2006.03.027.

Pinto, J. P., et al. (2014), Intercomparison of field measurements of nitrous acid (HONO) during the SHARP campaign, J. Geophys. Res. Atmos., 119, 5583–5601, doi: 10.1002/2013JD020287.

Dunlea, E. J., S. C. Herndon, D. D. Nelson, R. M. Volkamer, F. San Martini, P. M. Sheehy, M. S. Zahniser, J. H. Shorter, J. C. Wormhoudt, B. K. Lamb, E. J. Allwine, J. S. Gaffney, N. A. Marley, M. Grutter, C. Marquez, S. Blanco, B. Cardenas, A. Retama, C. R. Ramos Villegas, C. E. Kolb, L. T. Molina and M. J. Molina (2007). "Evaluation of nitrogen dioxide chemiluminescence monitors in a polluted urban environment." Atmos. Chem. Phys. J1 - ACP 7(10): 2691-2704.

Madronich, S., Photodissociation in the atmosphere: 1. Actinic flux and the effect of ground reflections and clouds, J. Geophys. Res., 92, 9740–9752, 1987.

Tie, X., S. Madronich, S. Walters, R. Zhang, P. Racsh, and W. Collins, Effect of clouds on photolysis and oxidants in thetroposphere,J. Geophys. Res.,108(D20), 4642, doi:10.1029/2003JD003659, 2003.

Stutz, J., Alicke, B., and Neftel, A.: Nitrous acid formation in the urban atmosphere: Gradient measurements of NO2 and HONO over grass in Milan, Italy, J. Geophys. Res., 107, 8192, doi:10.1029/2001jd000390, 2002.

Whalley, L. K., Stone, D., George, I. J., Mertes, S., van Pinxteren, D., Tilgner, A., Herrmann, H., Evans, M. J., and Heard, D. E.: The influence of clouds on radical concentrations: observations and modelling studies of HOx during the Hill Cap Cloud Thuringia (HCCT) campaign in 2010, Atmos. Chem. Phys., 15, 3289-3301, https://doi.org/10.5194/acp-15-3289-2015, 2015
* * *

---

## Author Comment (AC1) · 8 Mar 2019

Thank you for your valuable comments. We have carefully gone through all the comments and suggestions. To address these issues, we have included additional information and calculation results. The structure of the manuscript was also rearranged to improve the readability and logical structure. Please find the point-by-point response and revised manuscipt in the supplement zip file.

Please also note the supplement to this comment:
https://www.atmos-chem-phys-discuss.net/acp-2018-996/acp-2018-996-AC1-supplement.zip

---

## Author Response (AR1)

**Response to editor:**

We have carefully gone through all the comments and suggestions raised by the two reviewers. To address these issues, we have included additional information and calculation results. The structure of the manuscript was also rearranged to improve the readability and logical structure. Please find the point-by-point response to the reviewers and the marked revised manuscript below:

**Response to reviewer #1**

**General comments**

The article "NH$_3$-promoted hydrolysis of NO$_2$ induces explosive growth in HONO" discussed the mechanism behind explosive HONO formation during field observation in a rural site in North China. In general, the phenomenon, confidence of related evidence were sufficient to show the role of NH$_3$ in HONO productions via heterogeneous reaction during fog/smoke events. The observation data was well linked to possible atmosphere processes, which might greatly promote the understanding of HONO sources and thus providing new insights of pollution control strategies for China. Yet, the authors should address several minor points to make the narrative as well as the deduction more convincible.

**Minor suggestions**

1. First, it is well known that the HONO is extremely reactive especially during daytime. For most of the cases, the author noted that the rarely seen Ozone was the evidence that there was no sufficient sunlight. It seems that the Ozone concentration was used as an indicator of UV radiation and possible photochemistry reactions (line213-line218). But the Ozone could be titrated by NO, which was often measured a high level during nighttime in North China. Therefore, even the Ozone was observed to be nearly zero, there might be enough UV radiation for the quick HONO photolysis. This leads to a further question - can we trust the HONO measurements by a denuder system? The annular denuder method of detecting might have artefacts regarding to measuring HONO due to: 1. Hydrolysis of NO2 onto wet surface; 2.Aqueous reaction of S(IV) with NO2 in the solution(Spindler, Hesper et al. 2003, Nie, Ding et al. 2015). The second reaction could likely be accelerated in the presence of ammonia as reported in the previous studies (Cheng, Zheng et al. 2016, Wang, Zhang et al. 2016). Therefore, it is strongly recommended that the authors should conducted some validation of the HONO data from IGAC, given the fact that the HONO data obtained by denuder system need further calculation/reprocessing.

**Response:**

Thank you for your valuable comments. During the campaign in 2016, the IGAC instrument was borrowed from the Fortelice International Company. Unfortunately, circumstances do not allow us to borrow the instrument again for additional experiments, but we hope to prove the instrument reliable using the entire measurement dataset.

As already pointed out in your comment, instruments using wet denuders to collect gaseous HONO can cause sampling artefacts mainly via two pathways: 1) the $NO_2$ conversion on the surface of the sampling tube and the wet denuder and 2) the reaction of $NO_2$ with S(IV) in the absorption solution in wet denuder (Nie et al., 2015). The second pathway is avoided in IGAC by using a dilute ($5 \times 10^{-3}$ M) $H_2O_2$ solution, which quickly converts S(IV) to S(VI). The first artefact is often corrected for using a linear correction using slopes of 0.83-0.85. (Su, 2008;Qiang et al., 2014;Nie et al., 2015). Qiang et al. (2014) compared HONO measurements by an instrument called GAC-IC with that of LOPAP and found generally good agreement between both instruments after using a linear correction. Note that such linear adjustments do not alter the overall variation characteristics of HONO. The GAC and MARGA systems all consist of horizontally positioned wet denuders, in which the absorption solution might accumulate and cause additional artifacts. The IGAC system uses a vertically installed wet denuder, guaranteeing for the smooth outflow of the absorption solution. Overall, it is reasonable to believe that IGAC is able to capture the variation characteristics of HONO, even if a slope of 0.83 were used to correct the HONO data, the peaks would still reach 8.8, 7.9, 9.5 and 14.6 ppb, which is still very high.

Further, to prove that the observed peaks were not caused by instrument sampling artefacts, we analyzed the variation of observed HONO with $SO_2$, $NO_2$ and $NH_3$ during 15th Oct. to 25th Nov. 2016 (Fig.1). High HONO concentrations were typically observed under low $SO_2$ conditions, which proves that the sampling artefact due to the reaction of S(IV) and $NO_2$ in the wet denuder could be neglected. If the instrument would cause sampling artefacts due to $NO_2$ conversions, the high HONO concentrations should have been frequently observed under high $NO_2$ concentrations, which was not the case. The $NO_2$ concentrations at the occurrence time of the 4 peaks were all below 50 ppb. $NO_2$ often exceeded 50 ppb during the campaign, however, HONO stayed below 7 ppb throughout the whole campaign, except for the 4 cases studied in this work.

[Figure]

**Figure 1** Variation of HONO with $SO_2$ (y-axis), $NO_2$ (x-axis) and $NH_3$ (z-axis) during 15th Oct. to 25th Nov. 2016, with the large dots indicating the data points with HONO exceeding 7 ppb

2. The second point will be lying in the mechanism discussions. Though R1 could not explain the increasing nitrate (line 260-261), but attributing all the SIA (secondary inorganic aerosols) increase due to HONO formation and thus denying the role of R1 seems to be assertive. Actually, the nitrite in the aqueous phase might have produced OH radicals in aerosol liquid water or fog droplets (Vione, Maurino et al. 2006). It would be good to illustrate or maybe quantify the relative contribution of R1 v.s. R2 to HONO production as well as SIA production.

**Response:**

We thank the reviewer for these valuable comments and suggestions, which has greatly helped us in improving our manuscript. We took the advice in this comment and estimated the relative contributions of R1 and R2 to HONO production using the following assumptions.

First, we assume the observed increase in sulfate (d[SVI]/dt) was caused by the reaction of $SO_2$ with $H_2O_2$, $O_3$, $NO_2$, TMI ($Fe^{3+}$ and $Mn^{2+}$). Calculations were performed according to Cheng et al. (2016a), using the same pH dependent TMI concentrations and the actual $SO_2$, $H_2O_2$, $O_3$ and $NO_2$ concentrations in our measurements (Table 1). For the two fog episodes on 4[th] and 5[th] Nov. 2016, the mean diameter of fog droplets was assumed to be 7.0 μm and the liquid water content was assumed to be 0.3 g m$^{-3}$ according to Shen et al. (2018). For the haze episodes on the 11[th] and 14[th] Nov. 2016, the mean aerosol diameter under ambient conditions was estimated to be 0.65-1.22 and 0.9 μm, while the liquid water content was calculated to decrease from $3.4 \times 10^{-4}$ to $7.8 \times 10^{-5}$ on the 11[th] Nov and assumed to be 0.01 g m$^{-3}$ on the 14[th] Nov. during the transition from fog to haze. The sulfate production rate and relative contribution of the each oxidation pathway to the total sulfate production rate was obtained and depicted in Figure 2. For the two fog episodes, assuming pH=6, the estimated average sulfate production rates are 11.7 and 31.6 approximately 4 times of that observed within PM2.5, which is clearly an underestimation, considering the liquid water content of fog droplets are at least a magnitude higher than that of aerosols. For the two haze episodes, using the pH values estimated using ISORROPIA (forward mode and metastable assumption (Song et al., 2018)), the estimated average sulfate production rates are 0.06 and 1.8, about 10% of that observed within PM2.5. Following the calculations of Cheng et al. (2016a), we have considered the influence of ionic strength on the reaction rates and set constraints on the maximum ionic strength ($I_{max}$), which might have caused underestimations for all reaction routes, since the calculated ionic strength commonly exceeded $I_{max}$. Underestimated transition metal ion concentrations may also be partly responsible for the underpredicted sulfate production, since the TMI catalysis route has recently be pointed out to be the dominant $SO_2$ heterogeneous oxidation pathway (Shao et al., 2019). Additionally, there also might be other neglected $SO_2$ oxidation pathways, which will lead to overestimates in the sulfate fraction produced by the $NO_2$ oxidation pathway.:

$$\frac{d[HONO]}{dt}\bigg|_{R1} = 2 \times frac_{SO_2+NO_2} \times \frac{d[SVI]}{dt}\bigg|_{obs}. \tag{1}$$

where $frac_{SO_2+NO_2}$ is the contribution fraction of the $NO_2$ oxidation pathway to the total sulfate production. Note that the calculated HONO production rate can only represent the production within $PM_{2.5}$.

Table 1. The trace gas concentrations, liquid water content, mean diameter and temperature used to calculate the heterogeneous sulfate production

| Date | Time (LT) | $SO_2$ (ppb) | $H_2O_2$ (ppb) | $NO_2$ (ppb) | $O_3$ (ppb) | LWC (g m$^{-3}$) | $D_p$ (μm) | T (K) |
|---|---|---|---|---|---|---|---|---|
| 4$^{th}$ Nov | 9:00 | 0.18 | 0.26 | 45.3 | 1.53 | 0.3 | 7.00 | 277.8 |
| | 10:00 | 0.17 | 0.29 | 48.8 | 1.56 | 0.3 | 7.00 | 278.4 |
| | 11:00 | 0.28 | 0.34 | 49.9 | 1.78 | 0.3 | 7.00 | 278.7 |
| 5$^{th}$ Nov. | 10:00 | 0.16 | 0.19 | 44.6 | 2.90 | 0.3 | 7.00 | 278.8 |
| | 11:00 | 0.39 | 0.21 | 44.0 | 3.39 | 0.3 | 7.00 | 279.6 |
| | 12:00 | 1.19 | 0.30 | 45.1 | 5.72 | 0.3 | 7.00 | 281.3 |
| 11$^{th}$ Nov. | 7:00 | 0.40 | 0.52 | 30.7 | 1.41 | 3.4 e$^{-4}$ | 1.22 | 271.2 |
| | 8:00 | 0.44 | 0.71 | 33.0 | 1.53 | 2.1e$^{-4}$ | 0.73 | 272.3 |
| | 9:00 | 1.61 | 0.89 | 32.7 | 1.83 | 7.8e$^{-5}$ | 0.65 | 274.8 |
| 14$^{th}$ Nov. | 11:00 | 1.27 | 0.32 | 41.6 | 1.52 | 0.01 | 0.90 | 278.1 |

[Figure]

**Figure 2.** Calculated average sulfate production (a,c) and contribution fraction b,d) from SO$_2$ oxidation by H$_2$O$_2$, NO$_2$, O$_3$, TMI under different pH values using methods described in (Cheng et al., 2016a) for the case episodes on 4$^{th}$, 5$^{th}$, 11$^{th}$ and 14$^{th}$ Nov. 2016.

Then, we further assume the observed nitrate production (d[NO$_3^-$]/dt) was caused by reaction R2 and by the reaction of NO$_2$ with OH radicals ($k_{NO_2+OH}$=3.2×10$^{-12}$ cm$^3$ s$^{-1}$), the HONO production rate of R2 would be:

$$\frac{d[HONO]}{dt}_{R2} = \frac{d[NO_3^-]}{dt}_{obs} - k_{NO_2+OH}[NO_2][OH].$$ (2)

The contribution fraction of the two reactions to the heterogeneous HONO production in aerosol and fog liquid water content can be calculated by:

$$f_{R1} = \frac{d[HONO]}{dt}_{R1} / \frac{d[HONO]}{dt}_{R1+R2} \quad \text{and} \quad f_{R2} = \frac{d[HONO]}{dt}_{R2} / \frac{d[HONO]}{dt}_{R1+R2}.$$

Assuming the pH of fog droplets falls within the range of 4 to 6, $f_{R2}$ was estimated to range from range from 75.5 to 99.5% and from 81.2 to 99.5% during the 4$^{th}$ and 5$^{th}$ Nov. 2016, respectively. For the two haze events on 11$^{th}$ and 14$^{th}$ Nov., the $f_{R2}$ corresponding to the pH values modelled by ISORROPIA would be 98.2% and 97.3%.

These results suggest that, reaction R2 is the dominant contributor to the heterogeneous HONO production, while R1 is more important under high pH conditions. Under the assumed upper limit of pH, R1 can contribute up to 24.5%, 18.8% to the observed HONO growth during the fog events. This is in accordance with results from Wang et al. (2016) and Cheng et al. (2016b), which suggested that R1 was more likely to happen during fog episodes or under $NH_3$ neutralized conditions (3,4). For the two haze events, R1 contributed very little (1.8% and 2.7%) to the observed HONO growth.

In summary, reaction R2 was the dominant contributor to the heterogeneous HONO production, while R1 only played a minor role during fog events and a negligible role during haze events. The above discussions were added to Sect. 4.2 in the revised manuscript.

**Technical notes**

1. Line 110-111: "Under highly polluted conditions such as our site". Might have wrong grammar used.

**Response:**

Thank you for noticing, this sentence was rephrased as:

"Considering the severe pollution state the NCP is under, these measurement uncertainties are fully acceptable."

2. Figure 3. The time label on X axis causes misunderstanding, might change to Date-Time format.

**Response:**

Thank you for the suggestion, to avoid confusion the x axis labels were changed to hours and the date was marked on top of the figure (see Fig.4 below in Response #3).

3. Figure 4. The unit of aerosol composition (nitrate/sulfate/ammonium) should be in mass concentration.

**Response:**

Thank you for the suggestion, the unit of aerosol composition was changed to $\mu g\ m^{-3}$ in Figs. 2-4 (Figs. 3-5 below) and in the corresponding texts.

[Figure]

**Figure 3**. Time series of ambient **a)** RH; **b)** HONO; **c)** sulfate, nitrate, ammonium; **d)** $NH_3$, $NO_3$ and $SO_2$ during the observation period.

[Figure]

**Figure 4** Time series of ambient **a)** RH,$O_3$, **b)**HONO, $NO_2^-$, **c)** $SO_4^{2-}$, $NO_3^-$, $NH_4^+$, **d)** $NH_3$, $NO_2$, $SO_2$, **e)** NO, $H_2O_2$, **f)** CO, wind speed and wind direction (colors of scatter points ) from 11-04 to 11-05. Gray shaded areas represent periods of rapid increase of HONO.

[Figure]

**Figure 5** Time series of ambient **a)** RH, SO$_2$, **b)** HONO, NO$_2^-$, H$_2$O$_2$, aerosol pH, **c)** SO$_4^{2-}$, NO$_3^-$, NH$_4^+$, **d)** NH$_3$, NO$_2$, O$_3$, **e)** NO, volume concentrations of PM$_{2.5}$ in dry state (V$_{dry}$), volume concentrations of liquid water (V$_w$), **f)** CO, wind speed and wind direction during **1)** 11$^{th}$ Nov. 2016 and **2)** 14$^{th}$ Nov. 2016. Gray shaded areas represents periods of rapid increase of HONO.

   On my side, considering the IGAC experimental device and condition of use, I especially worry about the use of "a dilute $H_2O_2$ solution to collect the gases". If one refer to Young et al, 2016 the "dilute solution" is a $5x10^{-3}$ M solution (why not mentioning the concentration in the experimental section?) which is used to "assure the oxidation of $SO_2$ to $SO_4^-$ and prevents microbial growth ". For me, it is highly probable that such a concentration of such a strong oxidation agent could induced artefacts in the HONO measurement: - in the absence of precursors, it may induce a negative artefact by oxidizing nitrites to nitrates but, on the contrary - in the presence of enough reduced nitrogenous species (such as ammonia) it may forms HONO. In this case this would both affect $NH_3$ and HONO measurements and would probably lead to a correlation between both species (if ammonia is in excess). Considering the poor level of details provided in the experimental section, the lack characterization experiments demonstrating the ability of IGAC to measure HONO

(especially in the presence of ammonia) and the strong suspicion of artefacts exactly relevant from the main paper conclusion, I strongly recommend to provide the experimental evidences that demonstrate the suitability of the measurement protocol for both $NH_3$ and HONO before considering any publication.

*Response:*

During the campaign in 2016, the IGAC instrument was borrowed from the Fortelice International Company. Unfortunately, circumstances do not allow us to borrow the instrument again for additional experiments, but we believe the entire dataset itself may be able to prove itself reliable.

1) **On the concern that the IGAC instrument showed "poor" performance in respect of ammonia measurements**, we would thank the reviewer for the careful inspection of our work and for raising this concern, which indeed needs to be addressed to solidify our work.

   To evaluate the $NH_3$ data quality measured by IGAC, we compared them against $NH_3$ measurements of an LGR economical ammonia analyzer (DLT-100, Los Gatos Research, USA). Note that the IGAC instrument, the SMPS+APS system, the humidified nephelometer system and the AL2021 $H_2O_2$ analyzer were housed in an air-conditioned container located on the northern edge of the Gucheng site, while the trace gas instruments (including $SO_2$, $NO_x$, CO and $NH_3$) that carried out long-term measurements were housed on the second floor of a two story building located on the southern edge of the Gucheng site. Details on the LGR $NH_3$ measurements can be found in Meng et al. (2018).

   In Young et al. (2016), IGAC showed marginally acceptable performance test result for $NH_3$, with the intercept of the linear fitting meeting the evaluation criteria, while the slope of 0.59 did not. They pointed out that the underestimation in IGAC $NH_3$ measurements was probably caused by losses on the sampling tube wall. This indicates that the $NH_3$ measured by IGAC are systematically lower, which high possibly has no great influence on the relative variation pattern of $NH_3$, which we are concerned of in this study. It should be noted that Young et al. (2016) performed

NH$_3$ measurements and validation in the concentration range of 0 to 16 ppb. Liu et al. (2017) performed NH$_3$ measurements with IGAC in urban Beijing, with NH$_3$ varying between 0 to 35 ppb, which is closer to the observed NH$_3$ range in our campaign. The measurements were validated against ISORROPIA II simulations and reached good agreement (R$^2$>0.9). Teng et al. (2017) observed that LGR measurements were far larger than those measured based on wet denuders (with a slope near 0.7) and suggested the overestimation of LGR was possibly caused by interference of water vapor. Overall, we believe that even though the IGAC system may not be as precise as other specific NH$_3$ analyzers, if it can capture the variation characteristics of NH$_3$, it would be enough to prove the proposed the HONO production pathway.

Overall, the comparison of the NH$_3$ measured with LGR against that of IGAC showed a slope of 0.91 (R=0.63) and an intercept of 6.86 (Fig.1a). If data measured under high RH conditions (RH≥80) were excluded, an obvious improvement in the comparison results would be achieved (Fig.1b), with a slope of 1.03 (R=0.74) and an intercept of 2.89. This indicates that, high discrepancies between the two instruments mostly occur at high relative humidity, where LGR measurements are significantly higher than those of IGAC. Since these discrepancies were linked to high RH conditions, it is more likely that they were caused by the LGR instrument that overestimates NH$_3$ due to absorption interference of water vapor. The variation of NH$_3$ measured by LGR and IGAC during the four episode cases is shown in Fig. 2. Although LGR showed higher NH$_3$ concentrations (especially during nighttime when fog prevailed and RH was near 100), the variation characteristics were the same between the two instruments, all displaying rapid increases during the explosive HONO formation events. Thus, we believe it is appropriate to use the IGAC measured NH$_3$ for our discussions.

[Figure]

**Figure 1** Comparison between NH$_{3,IGAC}$ and NH$_{3,LGR}$ using a) all measurement data and b) data associated with RH<80.

[Figure]

**Figure 2** Time series of NH$_{3,IGAC}$ (solid) and NH$_{3,LGR}$ (dashed) during a) 4[th] Nov., b) 5[th] Nov., c) 11[th] Nov. 2016 and d) 14[th] Nov. 2016. Gray shaded areas represents periods of rapid increase of HONO.

We hope that this gives the reviewer and readers more confidence in the measurement results of IGAC concerning NH$_3$. In the revised manuscript, the following was added to Sect. 2 to provide the readers with more insight into the $NH_3$ measurements and data quality:

"A comparison between $NH_3$ observed by IGAC and by an economical $NH_3$ analyser (LGR, DLT-100, details see Meng et al. (2018)) yielded an overall slope of 0.91 with R=0.63 (Fig.S1a). A better comparison result (slope of 1.03, R=0.74) would be obtained if data associated with RH≥80 were excluded (Fig. S1b). The overestimation of LGR instruments compared to denuder based instruments has also been reported in Teng et al. (2017), suggesting possible interference of water vapor on $NH_3$ measurements. As can be seen in Fig S2., both instruments captured the same the diurnal variation of $NH_3$ during the four case episodes in this study, which proves that the IGAC instrument was able to capture the overall variation trends of $NH_3$. Since both instruments have their uncertainties, we decided to use the $NH_3$ measured by the IGAC instrument for better consistency with the other data."

2) **On the concern, whether the IGAC instrument can accurately measure HONO,** the performance of IGAC in terms of HONO measurements has indeed not been validated against measurements from other instruments at present. A recent work compared WRF-CHEM simulated HONO against measurements by IGAC and found good agreement between the two of them (Feng et al., 2018). The IGAC instrument is not as widely used as the MARGA system, which shares similar design and principles as IGAC and has been often used to measure HONO in the past (Xie et al., 2015;Nie et al., 2015). Similar measurement systems have been widely applied to study the variation of HONO (Su et al., 2008;Yang et al., 2017;Gu et al., 2009;Qiang et al., 2014). Instruments using wet denuders to collect gaseous HONO can cause sampling artefacts mainly via two pathways: 1) the $NO_2$ conversion on the surface of the sampling tube and the wet denuder and 2) the reaction of $NO_2$ with S(IV) in the absorption solution in wet denuder (Nie et al., 2015). The second pathway is avoided in IGAC by using the dilute ($5x10^{-3}$ M) $H_2O_2$ solution, which quickly converts S(IV) to S(VI). The first artefact is often corrected for using a linear correction using slopes of 0.83-0.85. (Su, 2008;Qiang et al., 2014;Nie et al., 2015). Qiang et al. (2014) compared HONO measurements by an instrument called GAC-IC with that of LOPAP and found generally good agreement between both instruments after using a linear correction. Note that such linear adjustments do not alter the overall variation characteristics of HONO. The GAC and MARGA systems all consist of horizontally positioned wet denuders, in which the absorption solution might accumulate and cause additional artifacts. The IGAC system uses a vertically installed wet denuder, guaranteeing for the smooth outflow of the absorption solution. Overall, it is reasonable to believe that IGAC is able to capture the variation characteristics of HONO, even if a slope of 0.83 were used to correct the HONO data, the peaks would still reach 8.8, 7.9, 9.5 and 14.6 ppb, which is still very high.

To further prove that the observed peaks were not caused by instrument sampling artefacts, we analyzed the variation of observed HONO with $SO_2$, $NO_2$ and $NH_3$ during $15^{th}$ Oct. to $25^{th}$ Nov. 2016 (Fig.3). High HONO concentrations were typically observed under low $SO_2$ conditions, which proves that the sampling artefact due to the reaction of S(IV) and $NO_2$ in the wet denuder could be neglected. If the instrument would cause sampling artefacts due to $NO_2$ conversions, the high HONO concentrations should have been frequently observed under high $NO_2$ concentrations, which was not the case. The $NO_2$ concentrations at the occurrence time of the 4 peaks were all below 50 ppb. $NO_2$ often exceeded 50 ppb during the campaign, however, HONO stayed below 7 ppb throughout the whole campaign, except for the 4 cases studied in this work.

[Figure]

**Figure 3** Variation of HONO with $SO_2$ (y-axis), $NO_2$ (x-axis) and $NH_3$ (z-axis) during 15th Oct. to 25th Nov. 2016, with the large dots indicating the data points with HONO exceeding 7 ppb

2. Line 213-214: "The $O_3$ concentration stayed near zero, which means that UV radiation was weak." This statement is clearly wrong. From the few NO data that the author disclose to the reader one can see that NO values are typically ranging from 20 to 100 ppb. With such high values, no wonder why $O_3$ remain low: it is clearly titrated by NO. One can understand that the lack of spectral radiometer measurements is an issue (see later) but $O_3$ data in a polluted environment can certainly not be used as a proxy for UV radiation strength.

*Response:*

We thank the reviewer for this valuable comment and agree that using $O_3$ as an indicator for UV radiation under such conditions is indeed not appropriate. As suggested by the reviewer in the following major comment, we used TUV calculation results to prove our point. During the case on the 14th Nov. 2016, the relative humidity decreased from 100% (10:00-11:00) to 85% (11:30), suggesting that this was a fog dissipation process. During 10:00 to 11:00, Gucheng was still under foggy conditions, with an estimated HONO lifetime (only considering its photolysis process) of 1.7 hours, proving that the photolysis process was relatively weak during the rapid increase of HONO. The estimated HONO lifetime rapidly decreased to 0.6 h by 12:00, resulting in accelerated HONO dissociation and OH production, explaining the rapid decrease of HONO concentrations.

The discussion in Line 213-214 was deleted and detailed discussions on the HONO photolysis were added in both Sect. 4.1 and Sect. 4.3 of the revised manuscript.

3. Line 226: Equation 1 is strongly oversimplified. On the HONO sinks side one clearly miss - photolysis that can certainly not be neglected. Even if radiation measurements are not available, the authors manage, later on in the paper, to evaluate some values that could be used here. Another approach would be to provide an upper limit evaluating the J value above haze using TUV for example (see Madronich et al, 1988 and Tie et al, 2003) - deposition can be taken into account by using as deposition velocity the value given by Stutz et al., 2002, for example. In addition in the presence of hydrometeors, one clearly miss the loss processes onto/into haze droplets. On the HONO source side, may well identified processes are missing such as direct emission and heterogeneous HONO formation from conversion of $NO_2$ on ground surface and aerosol surfaces.

*Response:*

We thank the reviewer for the valuable suggestions in this comment. However, we believe there has been a little misunderstanding, which needs to be clarified. In this section, we tried to speculate if these observed HONO explosive growth events could have been caused by other known sources. In the second paragraph (line 224-236), we wanted to estimate if the homogenous oxidation of NO to HONO could be strong enough to produce such large amounts of HONO. Thus, Equation 1 was not meant to describe the net production of HONO based on all of its sources and sinks. It only describes the net HONO production via homogeneous gas phase reaction. The impact of direct emission is considered in the 3rd and 4th paragraph (line 233-251), which discuss the potential impact of vehicle and biomass burning emissions. The conversion of $NO_2$ on ground surface and aerosol surface is exactly what we are mainly discussing in the next paragraphs in this section (line 252-340). **We try to improve this section considering the reviewers suggestions and discuss all the known sources and sinks together to improve both the readability and scientific quality of the manuscript.** We calculated the following sources and sinks of HONO:

1) Gas phase homogeneous production of HONO:

$$P_{NO+OH}^{net} = k_{NO+OH}[NO][OH] - k_{HONO+OH}[HONO][OH], \qquad (1)$$

where diurnal variation of OH concentrations was inferred from Whalley et al. (2015), replacing OH under fog conditions with $1 \times 10^5$ cm$^{-3}$).

2) Vehicle emissions: $P_{emi}$=Emission factor*[NOx]$_{vehicle}$, where the emission factor was assumed to be 1% (maximum emission factor of 0.8% used in Huang et al. (2017)) and the total observed NOx was attributed to vehicle emission to obtain an upper limit for the vehicle emission.

3)     Heterogeneous conversion on aerosol and ground surface:

Typically, the conversion of $NO_2$ on aerosol and ground surface is parameterized as a linear function of $NO_2$ uptake coefficients and surface to volume ratios (surface area densities) (Xue et al., 2014;Li et al., 2018):

$$P_{het}=(k_g+k_a)\times[NO_2], \qquad (2\text{-}1)$$

Ground: $k_g = \frac{1}{8} \cdot \vartheta_{NO2} \cdot \gamma_g \cdot \frac{S}{V}$    (2-2)

Aerosol: $k_a = \frac{1}{4} \cdot \vartheta_{NO2} \cdot \gamma_a \cdot S_a$    (2-3)

$\vartheta_{NO2}$ stands for the mean molecular speed, $\gamma_g$ and $\gamma_a$ for the uptake coefficient on ground and aerosol surface, S/V for the surface to volume ratio and Sa for the ambient aerosol surface area density. We estimate ground surface HONO production using a $\gamma_g$ of $1e^{-6}$ during night time and $2e^{-5}$ during daytime and an S/V of $0.1 \ m^{-1}$. The heterogeneous HONO production in aerosol and fog droplets were already calculated using a $\gamma_a$ range of $1e^{-4}$ to $1e^{-3}$ as suggested by Li et al. (2018). Since the surface area density under fog conditions were not measured, we can only estimate that dHONO/dt during fog events would exceed 40 ppb/hour based on calculation results in Fig.S1. For non-fog conditions, we used the ambient aerosol surface area density calculated using the humidified nephelometer and $\gamma_a=1e^{-4}$ to further calculate the variation of the HONO production on aerosol surface.

4)     Loss through photolysis:

$$L_{pho}=J_{Hono}\times[HONO], \qquad (3)$$

where J_{HONO} was modelled using the TUV model, assuming AOD to vary with pH (see Table S1 in revised supplement).

Loss through dry deposition:

$$L_{dep}=v_{dep}/H\times[HONO], \qquad (4)$$

where the dry deposition rate $v_{dep}$ was assumed to be $0.3 \ cm \ s^{-1}$ according to Stutz et al. (2002) and the boundary layer height H was interpolated from ECMWF ERA-interim data ( http://apps.ecmwf.int/datasets/data/interim-full-daily/).

Finally, the net production rate can be expressed as:

$$P_{HONO}^{net} = P_{NO+OH}^{net} + P_{het} + P_{emi} - L_{pho} - L_{dep} \qquad (6)$$

Fig. 4 displays the estimated production and loss of HONO via various routes, as well as the calculated and actually observed dHONO/dt during the 4th, 5th, 11th and 14th Nov. 2016. The estimated upper limit for vehicle emissions displays little variability during the day, with slight decreasing trends during the four events, proving that the observed HONO production could not have been caused by direct vehicle emissions. The net gaseous phase production of HONO ($P_{hom}^{net}$) contributed 0.15-0.18, 0.04-0.07, 0.27-1.04 and 0.25-1.53 ppb h$^{-1}$ during the 4 case events, displaying little influence during fog events and more during haze events. However, the estimated $P_{hom}^{net}$ was far from sufficient to explain the observed d[HONO]/dt. Dry deposition was typically high during the night within the shallow nocturnal boundary layer and decreased during the day with the increase of the boundary layer height. The calculated $L_{dep}$ contributed 0.5-0.9, 0.4-0.6, 2.7-4.3 and 0.05-0.3 ppb h$^{-1}$ to the loss of HONO. No significant decreases in $L_{dep}$ were observed during the two fog events, while increases were detected during the cases on 11th and 14th Nov. Not only was the variation in $L_{dep}$ unable to explain observed HONO productions, it further added to the discrepancy between observed and calculated d[HONO]/dt. During the four case events the $J_{HONO}$ respectively increased from $0.7\times10^{-4}$ to $2.5\times10^{-4}$ s$^{-1}$, $1.6\times10^{-4}$ to $2.4\times10^{-4}$ s$^{-1}$, $0.03\times10^{-4}$ to $1.4\times10^{-4}$ s$^{-1}$ and $1.6\times10^{-4}$ to $4.4\times10^{-4}$ s$^{-1}$, with $L_{pho}$ contributing 0.9-8.9, 2.2-7.8, 0.03-5.5 and 0.8-26.4 ppb h$^{-1}$ to the loss of HONO. $J_{HONO}$ increased significantly by the end of the HONO growth events to $2.9\times10^{-4}$, $4.3\times10^{-4}$, $2.6\times10^{-4}$ and $6.6\times10^{-4}$ s$^{-1}$, respectively, suggesting that the rapid drop of HONO concentrations was high probably caused by the rapid photolysis. Overall, $L_{pho}$ contributed most to the discrepancy between observed and calculated d[HONO]/dt.

Generally, the observed and calculated d[HONO]/dt agreed better with each other outside the HONO explosive growth periods, showing overestimations when aerosol liquid water contents were high, suggesting possible overestimation in the NO$_2$ uptake coefficient in the parameterization of $P_{het}$. This further suggests that the observed discrepancies in HONO production have mainly been caused by uncertainties in the heterogeneous formation estimates. The fact that HONO

drastically increased while NO₂ varied little (9:30 to 11:30, 5th Nov. and 6:30 to 8:30, 11th Nov.) or hardly increased even under drastic increases of NO₂ (8:30 to 11:30, 14th Nov.), but displayed explosive growth with increasing NH₃, could not be explained by current known HONO sources (direct emission or gas phase reactions). Additionally, these rapid increasing HONO phenomena were all observed under foggy or high RH conditions, which further affirms the suspicion that the HONO increase was caused by heterogeneous conversion of NO₂.

The above results were added to the discussions in Sect. 4.1.

[Figure]

**Figure 4** Estimated HONO emission from vehicles (blue), gas phase production (green), production on ground (orange) and aerosol surface (pink), loss through photolysis (yellow) and dry deposition (purple), as well as the calculated (dotted black) and actually observed (solid black) d[HONO]/dt on a) 4th, b) 5th, c) 11th and d) 14th Nov. 2016

4. Line 304-306 then line 326-329: In these section the photolysis of HONO is described being rapid (which is probably true) while it has been neglected earlier. I think the manuscript need reorganization to discuss more coherently the photochemistry of HONO under these conditions.

*Response:*

We thank the reviewer for pointing that out. Sect. 4.1 has been reorganized and new results (already mentioned in previous comment) were added. Discussion on the HONO photolysis were also made for the other three cases in the revised manuscript in Sect.4.1: "During the four case events the $J_{HONO}$ respectively increased from $0.7\times10^{-4}$ to $2.5\times10^{-4}\,\mathrm{s^{-1}}$, $1.6\times10^{-4}$ to $2.4\times10^{-4}\,\mathrm{s^{-1}}$, $0.03\times10^{-4}$ to $1.4\times10^{-4}\,\mathrm{s^{-1}}$ and $1.6\times10^{-4}$ to $4.4\times10^{-4}\,\mathrm{s^{-1}}$, with $L_{pho}$ contributing 0.9-8.9, 2.2-7.8, 0.03-5.5 and 0.8-26.4 ppb $\mathrm{h^{-1}}$ to the loss of HONO. $J_{HONO}$ increased significantly by the end of the HONO growth events to $2.9\times10^{-4}$, $4.3\times10^{-4}$, $2.6\times10^{-4}$ and $6.6\times10^{-4}\,\mathrm{s^{-1}}$, respectively, suggesting that the rapid drop of HONO concentrations was high probably caused by the rapid photolysis. Overall, $L_{pho}$ contributed most to the discrepancy between observed and calculated d[HONO]/dt."

**Minor issues**

1. Figure 2: it is somewhat disturbing that the figure does not displayed all the data acquired. In particular (but not only) the absence of NO and ozone data is clearly a problem. Furthermore, the use of "ppb" for aerosol composition is confusing: is it related to the whole volume of air? Is it related to the whole aerosol quantity. Please use more straightforward units here.

*Response:*

We included NO and $O_3$ concentrations in Figure 2 revised manuscript (see Fig. 5 below) according to the reviewer's suggestion. We also agree with the reviewer that the use of "ppb" as a unit for aerosol composition is confusing. The units in the text and in Figs. 2-4 in the revised manuscript were changed to "$\mu g\ m^{-3}$" instead (see Figs. 5-7 below).

[Figure]

**Figure 5**. Time series of ambient **a)** RH; **b)** HONO; **c)** sulfate, nitrate, ammonium; **d)** $NH_3$, $NO_3$ and $SO_2$ during the observation period.

[Figure]

**Figure 6** Time series of ambient **a)** RH,O₃, **b)**HONO, NO₂⁻, **c)** SO₄²⁻, NO₃⁻, NH₄⁺, **d)** NH₃, NO₂, SO₂, **e)** NO, H₂O₂, **f)** CO, wind speed and wind direction (colors of scatter points ) from 11-04 to 11-05. Gray shaded areas represent periods of rapid increase of HONO.

[Figure]

**Figure 7** Time series of ambient **a)** RH, SO₂, **b)** HONO, NO₂⁻, H₂O₂, aerosol pH, **c)** SO₄²⁻, NO₃⁻, NH₄⁺,**d)** NH₃, NO₂, O₃, **e)** NO, volume concentrations of PM₂.₅ in dry state (V_dry), volume concentrations of liquid water (V_w), **f)** CO, wind speed and wind direction during **1)** 11ᵗʰ Nov. 2016 and **2)** 14ᵗʰ Nov. 2016. Gray shaded areas represents periods of rapid increase of HONO.

2. Line 98 – 108: The experimental description of the instrument, the inlet and the protocol is insufficient.

*Response:*

We thank the reviewer for pointing it out and added more details to the experimental description on the instrument.

This part was revised as follows:

"During this field campaign, an In situ Gas and Aerosol Compositions Monitor (IGAC, Fortelice International Co.,Taiwan) was used for monitoring water-soluble ions (Na⁺, K⁺, Ca²⁺, Mg²⁺, NH₄⁺, SO₄²⁻, NO₃⁻,NO₂⁻, Cl⁻) of PM₂.₅ (particulate matter with aerodynamic diameter less than 2.5 μm) and trace gases including HONO, $SO_2$, $NH_3$, HCl, and $HNO_3$ with a time resolution of 1h. The IGAC system draws in ambient air through a PM10 inlet and passes the sample through a sharp-cut PM2.5 cyclone at a flowrate of 16.7 L/min. The total length of the stainless steel sampling line is approximately 2 m, with an inner diameter of 3.18 cm (1.25 inch), resulting in a residence time below 6 s, suggesting that underestimates in $NH_3$ possibly caused by adsorption on the stainless steel sampling tube as was proposed by Young et al. (2016) might be unimportant. A vertical annular denuder wetted with dilute $H_2O_2$ solution ($5 \times 10^{-3}$ M) collects the trace gases and converts $SO_2$ rapidly to $SO_4^{2-}$, preventing $SO_2$ from reacting with $NO_2$ in the absorption solution to produce HONO artefacts. A scrub and impact aerosol collector under the denuder is mounted at an inclined angle to capture particles based on impaction after condensation growth. Two separate Ion Chromatographs are used to respectively analyze anions and cations for the gas and aerosol liquid extracts which were injected from the denuder and the aerosol collector once an hour. The detection limits are below 0.12 μg/m$^3$ and the background concentration of most water-soluble inorganic ions within the instrument were below 0.11 μg/m$^3$, only with $SO_4^{2-}$ showing a background concentration of 1.10 μg/m$^3$ (Young et al., 2016). Under highly polluted conditions such as our site, these measurement uncertainties are fully acceptable. The instrument has shown good performance in the past, agreeing well with filter based samples (Liu et al., 2017). Standard LiBr solution was continuously added to the aerosol liquid extracts during the measurements, to ensure the sampling and analyzing process is stable. The swing amplitude was within the range of three standard deviation, confirming the stability of the ion analyzing system throughout the campaign. A mixed standard solution was diluted to perform multipoint calibrations (at 5, 10, 20, 50, 100, 200, 500 and 1000 ppb concentrations) at the beginning and at the end of the campaign for the ions $Na^+$, $K^+$, $Ca^{2+}$, $Mg^{2+}$, $NH_4^+$, $Li^+$, $SO_4^{2-}$, $NO_3^-$, $NO_2^-$, $Cl^-$, $Br^-$, with the $R^2$ of the calibrations reaching above 0.9999."

3. Line 120: The authors indicate the use of a NOx monitor 42CTL from thermo. It is not clear if this instrument is equipped with a Mo-based converter, a Blue light converter or both. In any case, the risk of interferences with HONO on the $NO_2$ and

NOx channels are high (through the conversion of HONO on heated Mo – see Dunlea et al, 2007 for example - or through its photolysis by the blue light. During some part of this field campaign the HONO values can be as high as 20 % of $NO_2$. In this case it would be necessary to evaluate the cross-sensitivities of $NO_2$ and HONO in the configuration of the chemiluminescence monitor used.

*Response:*

We thank the reviewer for making a good point. The TE-42CTL NOx monitor at the Gucheng site is only equipped with a Mo-based converter, which means that HONO, PAN and $HNO_3$ can interfere with the $NO_2$ and NOx measurements. This we will clarify in our revised manuscript. During the entire campaign, the median value of HONO/$NO_2$ reaches 6.8%, while 90% of the data display HONO/$NO_2$ below 12.7%. The largest HONO/$NO_2$ were observed during the explosive HONO growth episodes in this study, which are shown in Fig.3. The $NO_2$ data measured by the TE-42C (red line) is compared against that subtracted by HONO (yellow line). Assuming all the HONO were converted to NO by the Mo-based converter, the actual $NO_2$ concentration would be similar to the yellow line in Figure 3, which only during the rapid HONO increase shows relatively larger deviation from the red line. Although there is indeed an impact of HONO on $NO_2$ measurements, $NO_2$ concentrations subtracted by HONO are still in excess, not limiting its conversion to HONO.

[Figure]

Figure 8 HONO, $NO_2$, $NO_2$-HONO concentrations on a) 4th, b) 5th, c) 11th and d) 14th Nov. 2016

4. Line 127: "wavelength" is misspelled

*Response:*

Thanks, correction made in the revised manuscript.

5. Line 229: The value of $10^6$ radicals/cm$^3$ is taken as "typical for noontime haze condition" and later used in the equation 1. Even if the order of magnitude of this guess is probably not too wrong there is no reference provided. Furthermore, I don't think that the scientific community have the necessary background to raise a "typical value" for these quite peculiar conditions. I would rather recommend to refer to published work such as Whalley et al, 2015 (field) or Tie et al, 2003 (large scale modeling)

*Response:*

We thank the reviewer for the suggestion and have added the suggested reference by Whalley et al. (2015). The diurnal variation therein was used to estimate the diurnal variation of gas phase HONO production (see response to Major Comment #3).

6. Line 267-269: This statement is quite vague. Which anions are the authors referring to? More explanation are needed.

*Response:*

We added the following explanations to the introduction part, where this was first mentioned:

"Results of Yabushita et al. (2009) suggest that anions (such as Cl-, Br- and I-) greatly enhance the hydrolysis of $NO_2$ on water, and the $NO_2$ uptake coefficients of R2 can be enhanced several orders of magnitude by increasing electrolyte concentration."

[revised manuscript text omitted]

---

## Author Response (AR2)

**Response to editor:**

We carefully went through all the comments and suggestions raised by the third reviewer. Adjustments were made to the $NO_2$ data, to avoid interference from $NO_z$ species. A detailed comparison of observed $HONO/NO_2$ with past measurements was included to further prove that our HONO measurements were not systematically biased. A new $NO_2$ reactive uptake coefficient parameterization scheme was included in the budget analysis to also account for photo-enhanced $NO_2$ conversion. A detailed explanation on how $NH_3$ promotes the hydrolysis of $NO_2$ was provided for the reviewer in the response, while it was also further clarified in the revised manuscript. Please find the point-by-point response to the third reviewer and the marked revised manuscript below:

**Response to Reviewer #3**

**General comments:**
**Reviewer:** *I have mixed feeling about this paper. The authors propose a very interesting point that NH₃ may promote the heterogeneous production of HONO and thus positively affect the oxidation processes and secondary aerosol formation. If it was true, it provides important inputs for the understandings of the heterogeneous chemistry of NOx and the unknown sources of HONO. However, there are also several major issues regarding the measurement techniques, budget analysis and interpretation of the results. I would like to recommend that this manuscript can be considered for publication after the following concerns being properly addressed.*

**Response:**

First, we want to thank the reviewer for the careful inspection of our work and for all the valuable comments and suggestions. We fully understand why the reviewer may have mixed feelings and doubts about this paper, since the HONO measurements, upon which the entire work is based on, were made using a wet denuder system and only reached a 1-hourly time resolution. Initially, we also thought of these peaks as outliers caused by potential interference to the instrument. However, upon very careful and detailed inspection, we found solid evidence ruling out possible interferences and explaining these high HONO peaks, providing us with confidence that these data should not have been excluded. We hope our detailed response to the specific comments can boost your confidence in our work.

**Specific comments:**

1. *A major concern is about the measurement of HONO by the IGAC. To my experience, such wet denuder system usually tends to overestimate HONO compared to the other instruments such as LOPAP and DOAS. This may explain why the measured HONO/NO2 ratios in this study are so large, almost one order of magnitude higher than the other studies over the world. Furthermore, this HONO/NO2 ratio should be even underestimated in this study considering the potential overestimation of NO2 (I presume that the Thermo NO2 analyzer should use catalytic converter that may subject significant interference at non-urban sites). A rough check can be made by comparing the measured HONO+NO3-+NO2- against NO2. If catalytic converter was used, the measured "NO2"*

*should include NO2 and some NOz species including HONO, NO3- and NO2-, etc. For most cases, the explosive growth of HONO was dominated by only one data point (e.g. 4th and 14th November). Such results are in general possibly affected by the potential measurement interference for this data point. So the authors are encouraged to thoroughly evaluate the performance of this IGAC system for the HONO measurements. Additional inter-comparison experiments should be helpful for validating the IGAC measurements and exclude the potential interferences during the foggy and high RH conditions.*

**Response:**

We are grateful for these valuable comments and suggestions. We hope to provide the reviewer with some confidence in our measurements and shed some light on the understandable doubts on the HONO measurements in the following two points:

First of all, we thank the reviewer for the careful inspection and for pointing out that our $HONO/NO_2$ ratios were too large. We reinspected our manuscript and found that the reported average $HONO/NO_2$ ratio (0.18) was not consistent with reported average values of HONO and $NO_2$ concentration (3 ppb and 32 ppb, respectively). Then, we carefully went through our calculations again and detected an error in the $HONO/NO_2$ ratios given in Sect. 3 L152-153 caused by measurement unit conversion (HONO concentration was not converted from $\mu g \ m^{-3}$ to ppb). After correcting the error, the $HONO/NO_2$ ratio ranged from 0.018 to 0.6 with an average of 0.098.

The $NO_2$ instrument uses a Mo-based converter, interference of $NO_z$ species on $NO_2$ measurements do exist. Since particles were filtered out of the air sample before it entered the instrument, the interference should have mainly come from HONO, PAN and $HNO_3$ and no interference from $NO_3^-$ and $NO_2^-$ should have existed. PAN measurements were unfortunately not available at Gucheng. Past literature reported wintertime PAN in the North China Plain in the range of 0.23 to 3.51 ppb, with an average of 0.70 ppb. Compared to the measured $NO_2$ concentrations (ranging from 7.5 to 60.1 ppb with an average of 32.0 ppb), the influence of PAN should be minor. Due to the high concentrations in ammonia and the high relative humidity at our site, $HNO_3$ concentrations were typically low, falling in the range of 0.03 to 0.34 ppb, with an average of 0.13 ppb, suggesting that $HNO_3$ also interfered little with the $NO_2$ measurements. To account for the interference of HONO and $HNO_3$ on the $NO_2$ measurements, we replaced all the $NO_2$ in the manuscript with $NO_2^*$, which equals to the measured $NO_2$

subtracted by the measured HONO and $HNO_3$. Thus, we would arrive at an HONO/$NO_2^*$ range of 0.02 to 0.60, with an average of 0.11.

Accordingly, adjustments to the original sentence in Sect. 3 L152-153 were made based on the new calculation results:

**Original:** *The HONO/$NO_2$ ratio ranged from 0.03 to 0.75 with an average of 0.18, which is higher than the average HONO/NO2 ratio previously observed in China (Liu et al., 2014;Cui et al., 2018).*
**Revised:** The HONO/$NO_2^*$ ratio ranged from **0.018** to **0.60** with an average of **0.11**, which is moderately higher than the average HONO/$NO_2$ ratio previously observed in China (Liu et al., 2014; Cui et al., 2018).

The reason why the HONO/$NO_2^*$ in our work is moderately higher than those previously reported can be easily seen in Figure 1 below (Fig3 in revised manuscript), which displays the variation of HONO/$NO_2^*$ with PM2.5 mass concentration and ambient aerosol surface area ($S_A$). The variation of HONO/$NO_2$ with PM2.5 mass concentration from Cui et al. (2018) and with $S_A$ from Liu et al. (2014) are displayed in Fig. 2 for comparison. It can be discerned that HONO/$NO_2$ all reveal increases with PM2.5 and $S_A$, confirming that the conversion of $NO_2$ to HONO might be promoted by ambient aerosol loading. From the PM2.5 range in Fig.1a and Fig.2a it can be deduced that Gucheng was suffering from much heavier aerosol pollution. Cui et al. (2018) made HONO measurements using the LOPAP instrument and arrived at HONO/$NO_2$ ratios in the range of 0.0013 to 0.17, with an average of 0.062. For a PM2.5 range of 0 to 100 µg m$^{-3}$ (Fig.2a), the average HONO/$NO_2^*$ in our study increased from 0.06 to 0.07. Liu et al. (2014) made HONO measurements with a liquid coil scrubbing/UV−Vis instrument and reported that average HONO/$NO_2$ increased from 0.04 to 0.1 when $S_A$ increased from 200 to 1100 µm$^2$ cm$^{-3}$. For the same $S_A$ range, the average HONO/$NO_2$ in this study increased from 0.05 to 0.15, while the median value increased from 0.06 to 0.14. The above comparison suggests that our results are fully comparable to those reported in the past for China, which indicates that the IGAC system was not systematically overestimating in terms of HONO measurements.

[Figure]

**Figure 1** Boxplots displaying the variation of HONO/NO$_2^*$ with a) PM$_{2.5}$ concentration and b) ambient aerosol surface area density.

[Figure]

**Figure 2** Variation of HONO/NO$_2$ with a) PM2.5 mass concentration from Fig. 6 of Cui et al. (2018) and b) aerosol surface area from Fig. 1b in Liu et al. (2014)

Secondly, as we already explained in a previous response, the IGAC instrument was borrowed from the Fortelice International Company in 2016 for the measurement campaign. Unfortunately, circumstances do not allow us to borrow the instrument again for additional experiments. However, we believe there is strong evidence showing that that the high HONO peaks were not caused by instrument interference.

The performance of IGAC in terms of HONO measurements has indeed not been validated against measurements from other instruments (such as LOPAP and DOAS) at present. However, the MARGA system(Xie et al., 2015;Nie et al., 2015) or other wet denuder based instruments of similar design and principles (Su et al.,

2008;Yang et al., 2017;Gu et al., 2009;Qiang et al., 2014), have been often used to measure HONO in the past and have been validated against LOPAP. In HONO measurements, wet denuders cause sampling artefacts mainly via two pathways: 1) the $NO_2$ conversion on the surface of the sampling tube and the wet denuder and 2) the reaction of $NO_2$ with S(IV) in the absorption solution in wet denuder (Nie et al., 2015). The second pathway is avoided in IGAC by using the dilute ($5 \times 10^{-3}$ M) $H_2O_2$ solution, which quickly converts S(IV) to S(VI). The first artefact is often corrected for using a linear correction using slopes of 0.83-0.85. (Su, 2008;Qiang et al., 2014;Nie et al., 2015). Qiang et al. (2014) found generally good agreement between LOPAP and their wet denuder based instrument after using such a linear correction. Note that such linear adjustments do not alter the overall variation characteristics of HONO. Additionally, these wet denuder based systems all consist of horizontally positioned wet denuders, in which the absorption solution might accumulate and cause additional artifacts, whereas the IGAC system uses a vertically installed wet denuder, guaranteeing for the smooth outflow of the absorption solution. Overall, it is reasonable to believe that IGAC is able to capture the variation characteristics of HONO, even if a slope of 0.83 were used to correct the HONO data, the peaks would still reach 8.8, 7.9, 9.5 and 14.6 ppb, which is still very prominent.

To further elucidate that the observed peaks were not caused by instrument sampling artefacts, we analyzed the variation of observed HONO with $SO_2$, $NO_2$* and $NH_3$ during 15[th] Oct. to 25[th] Nov. 2016 (Fig.3). High HONO concentrations were typically observed under low $SO_2$ conditions, which proves that the sampling artefact due to the reaction of S(IV) and $NO_2$ in the wet denuder could be neglected. If the instrument would cause sampling artefacts due to $NO_2$ conversions on the sampling tube or in the wet denuder, the high HONO concentrations should have been frequently observed under high $NO_2$ concentrations, which was not the case. The $NO_2$* concentrations at the occurrence time of the 4 peaks were all below 35 ppb. $NO_2$ often exceeded 35 ppb during the campaign, however, HONO stayed below 7 ppb throughout the whole campaign, except for the 4 cases studied in this work.

[Figure]

**Figure 3** Variation of HONO with $SO_2$ (y-axis), $NO_2^*$(x-axis) and $NH_3$ (z-axis) during 15$^{th}$ Oct. to 25$^{th}$ Nov. 2016, with the large dots indicating the data points with HONO exceeding 7 ppb.

**2.** *Another concern is on the HONO budget analysis. 1)The Equation 1 was only valid under the assumption of steady state where production and loss of HONO were in balance. Would such an assumption stand in this study, especially for the morning period and foggy or high RH conditions? 2)The OH concentration is a key factor with large uncertainty in such a budget analysis. The diurnal variation of OH levels used for the calculation should be provided in the supplement. 3)It is not clear why the authors choose different uptake coefficients for the uptake of NO2 on ground (1\*10^-6 and 2\*10^-5) and aerosol surfaces (10^-4 to 10^-3). The uptake coefficients are highly uncertain and still under debate. How did the selections of OH and uptake coefficients affect the analysis results in this study? 4) Furthermore, large discrepancy exits between the calculated and observed d[HONO]/dt from Figure 5. What's the possible reason for such discrepancy?*
**Response:**

Thank you for your questions and suggestions, which promoted us to further improve the readability of our manuscript.

1) We believe there is some misunderstanding to be clarified about the budget analysis. We did not assume that the HONO production and loss were in balance, Equation 1 does not suggest that dHONO/dt=0, hence we do not make a steady state assumption. We are only listing the production and loss terms that contribute to dHONO/dt in this equation. By comparing the calculated HONO net production rate (dHONO/dt$_{cal}$) with the measured one (dHONO/dt$_{obs}$), we try to identify the sources that we have overestimated (leading to $P_{HONO}^{net}$ overestimations: dHONO/dt$_{cal}$>dHONO/dt$_{obs}$) and the ones we have are missing (leading to $P_{HONO}^{net}$ underestimations: dHONO/dt$_{cal}$<dHONO/dt$_{obs}$). However, due to the low time resolution, we might be underestimating the loss terms of HONO, which would only result in larger unknown sources of HONO during the 4 case episodes.

2) The OH levels used for calculations were added to the supplement as Figure S4:

[Figure]

**Figure S4.** OH assumption for HONO budget analysis based on Whalley et al. (2015), replacing OH under fog conditions with $1 \times 10^5$ cm$^{-3}$

3) The range for the NO$_2$ uptake coefficient on ground surface ($\gamma_g$) was assumed according to Li et al. (2010) and Xue et al. (2014). The reason why we used different uptake coefficients for aerosol surfaces ($\gamma_a$) is because results of Yabushita et al. (2009) suggested that anions (such as $Cl^-$, $Br^-$ and $I^-$) can greatly enhance the hydrolysis of $NO_2$ on water. By increasing electrolyte concentration, the $NO_2$ uptake coefficients can be enhanced by several orders of magnitude, with an estimated $\gamma_{max} \sim 2 \times 10^{-4}$. Yabushita et al. (2009) also summarized from past literature that the uptake coefficient ranged from $10^{-7}$ to $10^{-3}$. Since aerosols are rich in anions, especially $Cl^-$, Li et al. (2018b) adopted a $\gamma_a$ range of $10^{-4}$ to $10^{-3}$. More and more recent studies suggest that the missing daytime HONO source is connected to $NO_2$ conversion on aerosols (Huang et al., 2017;Liu et al., 2014). Since no measurements of fog droplet surface areas were made in this experiment, we used the $\gamma_a$ range as suggested by Li et al. (2018b) and a wide range of surface area densities to discuss the range of $NO_2$ conversion rate for both aerosol and fog conditions. This is where we found that this $\gamma_a$ range would produce large amounts of HONO during nighttime fog conditions, which was not observed, suggesting $\gamma_a$ to be overestimated at least during nighttime. For non-fog conditions, the ambient aerosol surface area density could be calculated and a $\gamma_a$ of $1 \times 10^{-4}$ was applied in the budget analysis, which still overestimated nighttime HONO production, however, could not explain the rapid growth of HONO during the morning.

To analyze the influences of the uptake coefficient and [OH] assumptions on the overall budget analysis, we have made some new calculations. Figure 4 shows the budget analysis without the impact of aerosol surface $NO_2$ conversions. You can see that compared to Fig.5 in the previous manuscript, the dHONO/dt agree better with each other outside the gray shaded areas, which marks the time range for the four cases in this study. This suggests again, that the $\gamma_a$ assumption was responsible for the overestimated HONO productions (dHONO/dt$_{cal}$>dHONO/dt$_{obs}$). At the same time, removing the aerosol surface HONO production resulted in more underestimated HONO production in the haze cases. This result indicates that the assumption of $\gamma_a = 1 \times 10^{-4}$ made in the previous manuscript was not appropriate, $\gamma_a$ should vary with time, it was overestimated during most of the time and underestimated during the 4 cases. Thus we made calculations by reducing nighttime $\gamma_a$ to $5 \times 10^{-6}$ (as in Li et al. (2010) and Xue et al. (2014)), enhancing daytime value to $2 \times 10^{-4}$ during daytime when solar radiation is below 400 Wm$^{-2}$ and to $2 \times 10^{-4}$x(solar radiation/400) for solar radiation above 400 Wm$^{-2}$.

Results are shown in Figure 5, which compared to Fig 5 in the previous manuscript displays better agreement between calculated and observed dHONO/dt during nighttime, however larger discrepancies during daytime, indicating that the variation of $\gamma_a$ was not necessarily connected to photo-enhancement, as was suggested in past studies (Li et al., 2010;George et al., 2005). For the haze case on 11$^{th}$ Nov., this alteration was still unable to explain the observed HONO production during the morning, however caused highly overestimated HONO production during the day.

[Figure]

**Figure 4** Same as Fig.5 in previous manuscript neglecting conversion of NO₂ on aerosol surface.

[Figure]

**Figure 5** Same as Fig.5 in previous manuscript using $\gamma_a = 5 \times 10^{-6}$) during night time, $2 \times 10^{-4}$ during daytime when solar radiation is below 400 $Wm^{-2}$, $2 \times 10^{-4} \times$(solar radiation/400) for solar radiation above 400 $Wm^{-2}$.

The impact of the OH assumption on the budget analysis can be seen in Fig. 6, where we increased the daytime OH concentration by 5 times, which would result in a maximum daytime OH concentration of 5.7 $\times 10^6$ molec $cm^{-3}$, which would be unrealistically high for wintertime foggy/hazy conditions. However, even under such high OH conditions, the unknown source of HONO during the four case events could not be explained.

[Figure]

**Figure 6** Same as Fig.5 in previous manuscript using 5 times the [OH] as assumed before.

4) The positive discrepancies in Figure 5 suggest that the uptake coefficient on aerosols might be overestimated for nighttime conditions, since we used a value close to the upper limit proposed by Yabushita et al. (2009). The large negative discrepancies between calculated and observed dHONO/dt have all occurred during the HONO explosive growth events, indicating that there are missing sources of HONO unexplained by current HONO production mechanisms. Response to 1) and 3) also prove that this missing source was not caused by the low time resolution of the HONO data or by the uncertainties in OH and uptake coefficient assumptions. The most recently proposed mechanism of $NH_3$-promoted $NO_2$-hydrolysis (Li et al., 2018a), which was not included in current mechanisms of $NO_2$-to-HONO conversion, and the coinciding simultaneously growing $NH_3$ concentrations (in two independent $NH_3$ measurements) during these four events gave us the only explanation on how these four rapid HONO growth events came to be.

**3.** *For Equation 8, I suggest to use the partitioning of HNO3 to the aerosol phase to instead the k[NO2][OH]. Using k[NO2][OH] may*

*underestimate the nitrate formation if the ambient HNO3 was abundant. I presume the measurement data of HNO3 should be available from the IGAC system.*

**Response:**
Thank you for this suggestion, which is very reasonable for conditions with abundant $HNO_3$. However, in our study, due to the high relative humidity and $NH_3$ concentrations, the $HNO_3$ was very low (0.03 to 0.34 ppb, with an average of 0.13 ppb), suggesting that most of the produced $HNO_3$ was rapidly partitioned into the aerosol phase. Even if we used $HNO_3$ measurements, we could not replace k[$NO_2$][OH], because the d[$HNO_3$]/dt is both influenced by the partitioning between aerosol/fog and gas phase, as well as its homogeneous formation process. If we only consider the partitioning, we would probably cause large underestimations in homogeneously formed nitrate.

To account for $HNO_3$, we changed Equation 8 to:

$$\frac{d[HONO]}{dt}\bigg|_{R2} = \frac{d[NO_3^- + HNO_3]}{dt}\bigg|_{obs} - k_{NO_2 + OH}[NO_2][OH].$$ (Eq.8)

Since $HNO_3$ concentrations were very low compared to $NO_3^-$, the influences were minimal.

**4.** *Section 4.2: could the authors comment on how NH3 involve in the heterogeneous reactions of NO2? Does particulate ammonium work in the same manner? Is there any other indication that NH3 promotes the heterogeneous HONO production other than the moderate correlation between HONO/NO2 and NH3?*

**Response:**
1) According to Li et al. (2018a), gaseous $NH_3$ can markedly lower the free-energy barrier to HONO formation by the hydrolysis of $NO_2$ dimers (Fig.4 in Li et al. (2018a)), which means that $NO_2$ dimers can more easily overcome the activation energy needed for the hydrolysis reaction and produce HONO. Without $NH_3$, the $NO_2$ hydrolysis is an endergonic reaction, more energy is needed for the reaction to occur and the products (HONO, nitrate and $NH_4^+$) are less stable. Adding $NH_3$ turns the reaction into an exergonic reaction, stabilizing the state of the reaction products, which leads to the result that the products are not prone to react backwards. Their metadynamics simulation results show that the existence of water droplet surface would further reduce the free-energy barrier to 0.5 kcal/mol (Fig.5 in

Li et al. (2018a)), which is negligible under room temperature.

[Figure]

**Fig. 4.** Relative free-energy variation ($\Delta G$) along the corresponding reaction coordinates obtained from the thermodynamic integration methods. For $NH_3$-containing ($N_2O_4 + H_2O + NH_3$ and $N_2O_4 + 4H_2O + NH_3$) and $NH_3$-free ($N_2O_4 + 2H_2O$ and $N_2O_4 + 5H_2O$) systems, the collective variable is scanned every 0.2 Å from 2.4 to 4.8 Å and every 0.1 Å from 1.4 to 2.5 Å, respectively. One or more scanning points are interpolated to locate the minimum or the maximum points. For each single point, the constraint BOMD simulation runs for 5 ps and the next 10- to 15-ps simulation results are used for the free-energy calculation. The free-energy difference is obtained via an integration of the average Lagrange multiplier that is the average force required to constrain the collective variable at the desired value.

[Figure]

**Fig. 5.** (*A*) Time evolution of the $N_1$–$O_1$, $O_1$–$H_1$, and $N_2$–$H_1$ lengths during the BOMD simulation of $N_2O_4$ + $NH_3$ on the water droplet surface. (*B*) Relative free-energy variation ($\Delta G$) along the corresponding reaction coordinates obtained from thermodynamic integration methods.

2) The proposed mechanism of Li et al. (2018a) is a surface reaction with gaseous $NH_3$, which produces particulate $NH_4^+$ (Fig.5 in Li et al. (2018a)), hence particulate ammonium cannot act the same way as gas phase $NH_3$ does.

3) As for the question whether there are other evidences proving that our observations are linked to the new proposed $NH_3$-promoted $NO_2$ hydrolysis: the HONO budget study already ruled out the possibility that other HONO sources (vehicle emission, homogenous production, etc.) could have contributed to such rapid HONO formation. The current parameterization of heterogenous $NO_2$ conversion was also unable to explain such phenomenon. Uptake coefficient assumptions can lead to large uncertainties. The variation of HONO/$NO_2$ ratio with $PM_{2.5}$ and $S_A$ is often used to prove that aerosol surface can promote $NO_2$ conversion to HONO, however, as can be seen in the boxplots in Fig.1 and Fig.2, the correlation between HONO/$NO_2^*$ ratio and $PM_{2.5}$ and $S_A$ bears large variability. Further, it can be noticed that for the relatively lower $PM_{2.5}$ concentration and $S_A$ range, HONO/$NO_2$ increased rapidly with increasing aerosol loading, while after a critical concentration ($PM_{2.5}$>225 µg m$^{-3}$, $S_A$>1100 µm$^2$ cm$^{-3}$) the increase came to a halt. This indicates that under relatively cleaner conditions, the heterogeneous conversion of $NO_2$ to HONO might be limited by aerosol surface area density. However, under severe haze pollution or foggy conditions with sufficient $S_A$ available for heterogeneous reactions, the HONO formation was not sensitive to the change in $S_A$ anymore. Current parametrization schemes all assume that the heterogeneous conversion rate of $NO_2$ to HONO is positively correlated to $S_A$, which is clearly not the case for severe haze or fog events.

In comparison, the correlation of HONO/$NO_2^*$ with $NH_3$ is much more significant (Fig.7 a), showing much smaller variability and clearly demonstrating that high HONO/$NO_2^*$ (above 0.2) do only occur under simultaneously high $NH_3$ and high relative humidity conditions, proving that the abundance of gaseous $NH_3$ and water surface was promoting the HONO formation rather than surface areas.

[Figure]

**Figure 7** The relationship between $NH_3$ concentration and a) HONO/$NO_2^*$ ratio; b) nitrate/nitrogen dioxide ratio ($NO_3^-$/$NO_2^*$); Colors of scatter points represent ambient RHs and the color bar is shown on the top.

**5.** *Lines 88-89: delete "for the first time" as wet denuder systems have been widely used in field studies.*

**Response:**

Thank you for this suggestion, "for the first time" has been removed from this sentence.

[revised manuscript text omitted]

---

## Author Response (AR3)

**Response to editor:**

We are thankful for the editing work of the editor and are glad that this paper was accepted for publication. Upon the stage of final draft submission, we wanted to let the editor know that during the second review process, we have made a small change to the calculation of the TMI catalyzed $SO_2$ oxidation process, so that the pH dependent Fe and Mn concentrations are in accordance with those assumed in Cheng et al. (2016). This has resulted in more TMI catalyzed sulfate production under lower pH values, however, had negligible influence on the our results and conclusions.

The estimated average sulfate production rates for the two haze periods increased to 0.33 and 0.94 $\mu g\ m^{-3}\ h^{-1}$ (about 38% and 20% of that observed within $PM_{2.5}$). While for the estimated HONO production fraction of R1 and R2, $f_{R2}$ was estimated to range from range from 82.2 to 99.7% and from 86.8 to 99.8% during the 4[th] and 5[th] Nov. 2016, respectively. For the two haze events on 11[th] and 14[th] Nov., $f_{R2}$ was estimated to be 99.7% and 98.0%.

Under the assumed upper limit of pH, R1 could have contribute up to 17.8% and 13.2% to the observed HONO growth during the two fog events. For the two haze events, R1 contributed very little (0.3% and 2%) to the observed HONO growth.

[Figure]

**Figure 1** Calculated average sulfate production (a,c) and contribution fraction b,d) from SO$_2$ oxidation by H$_2$O$_2$, NO$_2$$^*$, O$_3$, TMI under different pH values using methods described in (Cheng et al., 2016) for the case episodes on 4$^{th}$, 5$^{th}$, 11$^{th}$ and 14$^{th}$ Nov. 2016.